# Queue Up Your Regrets: Achieving the Dynamic Capacity Region of Multiplayer Bandits

**Ilai Bistritz, Nicholas Bambos**
Stanford University
{bistritz,bambos}@stanford.edu

## Abstract

Consider $N$ cooperative agents such that for $T$ turns, each agent $n$ takes an action $a_n$ and receives a stochastic reward $r_n(a_1, \ldots, a_N)$. Agents cannot observe the actions of other agents and do not know even their own reward function. The agents can communicate with their neighbors on a connected graph $G$ with diameter $d(G)$. We want each agent $n$ to achieve an expected average reward of at least $\lambda_n$ over time, for a given quality of service (QoS) vector $\boldsymbol{\lambda}$. A QoS vector $\boldsymbol{\lambda}$ is not necessarily achievable. By giving up on immediate reward, knowing that the other agents will compensate later, agents can improve their achievable capacity region. Our main observation is that the gap between $\lambda_n t$ and the accumulated reward of agent $n$, which we call the QoS regret, behaves like a queue. Inspired by this observation, we propose a distributed algorithm that aims to learn a max-weight matching of agents to actions. In each epoch, the algorithm employs a consensus phase where the agents agree on a certain weighted sum of rewards by communicating only $O(d(G))$ numbers every turn. Then, the algorithm uses distributed successive elimination on a random subset of action profiles to approximately maximize this weighted sum of rewards. We prove a bound on the accumulated sum of expected QoS regrets of all agents, that holds if $\boldsymbol{\lambda}$ is a safety margin $\varepsilon_T$ away from the boundary of the capacity region, where $\varepsilon_T \to 0$ as $T \to \infty$. This bound implies that, for large $T$, our algorithm can achieve any $\boldsymbol{\lambda}$ in the interior of the dynamic capacity region, while all agents are guaranteed an empirical average expected QoS regret of $\tilde{O}(1)$ over $t = 1, \ldots, T$ which never exceeds $\tilde{O}(\sqrt{t})$ for any $t$. We then extend our result to time-varying i.i.d. communication graphs.

## 1 Introduction

Consider a group of artificial cooperative agents interacting in a shared environment. Each agent has a reward $r_n(\boldsymbol{a})$ that is a function of the actions of all agents $\boldsymbol{a}$ and represents a local performance metric. Examples include wireless devices, robots in a facility, or autonomous vehicles. The designer wants to program these agents with a distributed protocol that guarantees at least a certain average reward for each agent, which we term its QoS. Wireless devices interact by transmitting and interfering with each other over multiple channels, and the wireless protocol guarantees the advertised throughput for all of them. Multitasking robots interact by cooperating to make progress on their tasks at a certain rate, following a distributed learning algorithm with this objective in mind. The main challenge in designing such a distributed protocol or algorithm is that the agents are local and do not know enough about each other to guarantee the QoS of their peers.

In data-driven decision-making, agents do not know their reward function, and can only observe its value at the current action profile $\boldsymbol{a}(t)$, known as bandit feedback. The agents also do not know the reward functions of their peers, or even their bandit feedback. As a result, agents have to learn both how to optimize their reward function, and when to prioritize other agents instead.

36th Conference on Neural Information Processing Systems (NeurIPS 2022).

The QoS is often measured over time. Throughput in communication networks is measured over seconds, and progress in a factory is measured over minutes. In resource allocation, the optimal policy is often dynamic and combines time-sharing and simultaneous access. For example, consider four agents and one resource, such that each agent can access the resource or not. If Agents 1 and 2 access the resource simultaneously, they each get zero reward, and the same is true for Agents 3 and 4. Otherwise, an agent gets a reward of 1 by accessing the resource. This simple example arises in many applications. In wireless networks, far away devices can share the same channel, but nearby devices have to pick different channels. In cloud computing, a job that requires little RAM can run simultaneously with a RAM-intensive one, but two RAM-intensive jobs cannot share the CPU simultaneously. Similarly, we cannot charge too many electric vehicles that share a transformer simultaneously. In all these cases, dynamic algorithms can achieve QoS vectors that static multiagent algorithms cannot, where agents learn when to access, but also, who to access with.

*The main objective of this paper is to design a distributed multiagent algorithm that can achieve any dynamically feasible QoS vectors based only on bandit feedback and minimal communication.*

## 1.1  Our Contributions

We define the QoS regret of agent $n$ by $R_n(t) = \max\left\{0, \lambda_n t - \sum_{\tau=1}^t r_n(\boldsymbol{a}(\tau))\right\}$, given a QoS vector $\boldsymbol{\lambda}$. **Our main observation is that $R_n(t)$ behaves like a queue**, which inspires an algorithm that learns to approximate a max-weight matching between agents to actions to guarantee the stability of these "queues". A perfect centralized max-weight policy [1] plays the $\boldsymbol{a}$ that maximizes $\sum_{n=1}^N b_n R_n(t) r_n(\boldsymbol{a})$ for some positive $\{b_n\}$. To be able to learn this policy distributedly, our algorithm divides the time horizon of length $T$ into epochs of length $E_T$. In each epoch, agents use simultaneous distributed averaging (i.e., consensus) steps to agree on $\sum_{n=1}^N b_n R_n(t) r_n(\boldsymbol{a}(t+1))$ from recent turns, on a row stochastic communication matrix $W$. Then, agents use a distributed and scalable successive elimination to approximately maximize $\sum_{n=1}^N b_n R_n(t) r_n(\boldsymbol{a})$. Our consensus step does not require a doubly stochastic $W$ and converges in finite time.

We show that for large $T$, our algorithm can achieve any $\boldsymbol{\lambda}$ in the interior of the dynamic capacity region, where all agents have an empirical average expected QoS regret of $\tilde{O}(1)$ over time which never exceeds $\tilde{O}(\sqrt{t})$ for any $t$. Our algorithm only requires each agent to communicate $O(d(G))$ numbers each turn, where $d(G)$ is the diameter of the communication graph $G$. Furthermore, for finite $T$, we bound the accumulated expected sum of QoS regrets as a function the epoch length $E$, and characterize the safety margin $\varepsilon_E$ away from the boundary of the capacity region that is required to make $\boldsymbol{\lambda}$ feasible. We then prove QoS regret guarantees for i.i.d. time-varying communication graphs, modifying both the bound on the accumulated expected sum of QoS regrets and $\varepsilon_E$. For this more complicated case, the communication overhead is $O(\log^2 E)$ per agent per turn.

## 1.2  Related Work

Our work brings together game theory, learning, and queuing theory. Aspects of learning in queuing were studied in [2–5]. Queuing games were studied in [6], where each agent chooses which queue to join. In contrast, our analysis uses the observation that the QoS regret behaves like a queue, but no real backlogs exist. We consider general discrete games that may have nothing to do with queues.

Multiplayer bandits is an emerging field that studies how agents can learn to achieve a common goal [7–11] given bandit feedback of their unknown reward functions. Most works on multiplayer bandits assume the collision model, in which when multiple agents choose the same arm, they all receive zero reward [12–17]. Instead of the sum of rewards, our objective is to guarantee that the empirical average reward of each agent is at least a certain QoS. Multiplayer QoS bandits were studied in [18], assuming the collision model. However, [18] only achieves QoS vectors $\boldsymbol{\lambda}$ for which there exists an $\boldsymbol{a}$ such that $\mathbb{E}\{r_n(\boldsymbol{a})\} \geq \lambda_n$ for all $n$. Compared to [18], our work considers a general discrete game. But even for the collision model, for the example on the top of this page, two agents get zero reward in **any** static solution like [18], while we can achieve $\lambda_n = 0.5$ for all $n$.

Dynamic multiplayer bandits, where agents can join at different times, were studied in [12, 15, 17] under the collision model. More related to our work, [19], and recently [20], combined queuing with multiplayer bandits. In [19, 20], each agent (queue) receives packets randomly, and each packet is

transmitted to a server of choice, where servers have constant [19] or agent-dependent [20] service rates. Each server can only process one received packet each turn, given a selection rule. In contrast, our scenario has no actual queues or packets and considers a general (discrete) game, so the "service rate" for each agent is an arbitrary function of the actions of all agents (i.e., its reward function).

In power control, [21] can achieve any target QoS vector that is achievable by a static transmission power profile. The interaction between the wireless links, through the interference, is a game with the transmission powers as the actions. From this point of view, our work generalizes [21] in two ways: from power control to a general game (but discrete), and from a static to a dynamic solution.

## 2    Problem Formulation

We consider a general multiplayer multi-armed bandit game. At every turn $t \geq 1$, for $T$ turns, each agent $n$ in the set $\mathcal{N} = \{1, ..., N\}$ picks an action $\boldsymbol{a}_n(t) \in \{0, \ldots, K-1\}$. The expected reward for agent $n$ is an unknown function of the actions of all agents $\mu_n : \{0, \ldots, K-1\}^N \to \mathbb{R}$. Each agent $n$ only observes a stochastic reward, $r_n(\boldsymbol{a}(t)) \in [0, 1]$, such that $\mu_n(\boldsymbol{a}) = \mathbb{E}\{r_n(\boldsymbol{a})\}$ and $z_n(t) \triangleq r_n(\boldsymbol{a}(t)) - \mu_n(\boldsymbol{a}(t))$ is independent over time and between agents. Agents only know the action they played and cannot observe the actions of others.

Our goal is to design a distributed algorithm for the cooperative agents, that dictates the actions $a_n(1), \ldots, a_n(T)$ of each agent $n$, such that ideally

$$\min_n \left( \frac{1}{T} \sum_{t=1}^{T} \mu_n(\boldsymbol{a}(t)) - \lambda_n \right) \geq 0 \tag{1}$$

i.e., the empirical average of the expected reward of agent $n$ is at least $\lambda_n$ for all $n$.

The objective in (1) is only possible for certain QoS vectors $\lambda_1, \ldots, \lambda_N$. The achievability region of the dynamic problem in (1) can be much larger than the static one which includes all $\boldsymbol{\lambda}$ for which $\max_{\boldsymbol{a}} \min_n (\mu_n(\boldsymbol{a}) - \lambda_n) \geq 0$. As an example, consider a game with $N = 2$ and $K = 2$, such that $\mu_1(1, 0) = 1$, $\mu_1(0, 0) = \mu_1(0, 1) = \mu_1(1, 1) = 0$ and $\mu_2(0, 1) = 1$, $\mu_2(0, 0) = \mu_2(1, 0) = \mu_2(1, 1) = 0$. Then $\lambda^* = \max_{\boldsymbol{a}} \min_n \mu_n(\boldsymbol{a}) = 0$ is obtained for any fixed $a_1, a_2$, and no $\lambda_1, \lambda_2 > \lambda^*$ are feasible. However, if the agents play $(1, 0)$ in even turns and $(0, 1)$ in odd ones then $\frac{1}{T}\sum_{t=1}^{T} \mu_1(\boldsymbol{a}(t)) = \frac{1}{T}\sum_{t=1}^{T} \mu_2(\boldsymbol{a}(t)) = \frac{1}{2} > \lambda^*$. This simple example arises often when time-sharing of resources is better than simultaneous access. In general, the optimal strategies can involve softer time-sharing or time-sharing between groups of agents, based on their characteristics.

This motivates defining the dynamic capacity region [1]:

**Definition 1.** Let $\varepsilon \geq 0$. The approximate dynamic capacity region of the game $\mathcal{G} = \left(\mathcal{N}, \{0, \ldots, K-1\}_{n \in \mathcal{N}}, \{\mu_n\}_{n \in \mathcal{N}}\right)$ is defined as the set:

$$C_\varepsilon(\mathcal{G}) = \left\{ (\lambda_1, \ldots, \lambda_N) \mid \exists \boldsymbol{p} \in \Delta^{K^N} \text{ s.t. } \sum_{\boldsymbol{a}} p(\boldsymbol{a}) \mu_n(\boldsymbol{a}) > \lambda_n + \varepsilon, \forall n \right\} \tag{2}$$

where $p(\boldsymbol{a})$ is the probability of $\boldsymbol{a}$ and $\Delta^{K^N}$ is the $K^N$-th dimensional simplex. The (exact) dynamic capacity region is then $C(\mathcal{G}) \triangleq C_0(\mathcal{G})$.

A learning algorithm that has no knowledge of the game will need time to achieve (1), so we can only hope to achieve (1) up to some $T$-dependent error, which we term the QoS regret.

**Definition 2.** The QoS regret of agent $n$ at turn $t$ is defined as

$$R_n(t) \triangleq \max \left\{ 0, \lambda_n t - \sum_{\tau=1}^{t} r_n(\boldsymbol{a}(\tau)) \right\}. \tag{3}$$

The expected QoS regret encodes that an agent "regrets" the total accumulated reward it misses to have an expected empirical reward of at least $\lambda_n$. On the other hand, an agent with an empirical average reward of more than $\lambda_n$ does not require anything more so it has zero QoS regret. Note that if the expected QoS regret of all agents is zero then (1) holds with probability 1.

Our contributions are a novel distributed algorithm that can learn to achieve a target QoS vector, with the following performance guarantees:

**Theorem 1.** *Let $\mathcal{G}$ be a multiplayer multi-armed bandit game, played for $T$ turns. Let the agents run Algorithm 1 on a connected communication graph $G$, with epoch length $E_T$ such that $E_T \to \infty$ and $\frac{E_T}{T} \to 0$ as $T \to \infty$. If $\boldsymbol{\lambda} \in C(\mathcal{G})$, then for large enough $T$, we have $\frac{1}{T} \sum_{t=0}^{T-1} \mathbb{E}\{R_n(t)\} = O(E_T)$ for all $n$ and $\mathbb{E}\{R_n(t)\} = O\left(\sqrt{tE_T}\right)$ for all $n$ and $t$, where the expectation is with respect to the randomness of the algorithm and the rewards. To achieve this, each agent only needs to communicate $O(d(G))$ numbers[1] per turn, where $d(G)$ is the diameter of $G$.*

The QoS regret guarantees of Theorem 1 are asymptotic in $T$. For finite $T$, a tradeoff arises between $\mathbb{E}\{R_n(T)\}$ and the safety margin $\varepsilon_T$ we have to take from the boundary of $C(\mathcal{G})$ to guarantee this $\mathbb{E}\{R_n(T)\}$. Theorem 2 in the next section analyzes this tradeoff explicitly and shows that $\varepsilon_T \to 0$ as $T \to \infty$, which implies Theorem 1 above as a special case. By tuning the epoch length $E$, we can make $\varepsilon_T$ smaller for finite $T$, at the price of increasing the expected QoS regret, or vice versa. As long as $E_T$ increases with $T$, then $\varepsilon_T \to 0$ and our algorithm can achieve any $\boldsymbol{\lambda} \in C(\mathcal{G})$ for large enough $T$. Hence, if $E_T = \log T$ (or slower), then Theorem 1 implies that the empirical average expected QoS regret of all agents is $\tilde{O}(1)$ and no expected QoS regret exceeds $\tilde{O}(\sqrt{t})$ for any $t$. For finite $E$, these two bounds slightly improve to $O(1)$ and $O(\sqrt{t})$ respectively, as long $\boldsymbol{\lambda} \in C_{\varepsilon_E}(\mathcal{G})$.

In Section 12 in the appendix, we generalize Theorem 1 for a stochastic time-varying i.i.d. communication graph where only the union of the support set is assumed to be connected. This requires some slight modifications to our algorithm, detailed in Algorithm 2, among them using the delay parameter $D_E = O(\log^2 E)$. As a result, each agent needs to communicate $O(\log^2 E)$ numbers per turn.

It might look tempting to view (1) as the two-player zero-sum game $\min_{\boldsymbol{p} \in \Delta^{K^N}} \max_n \left(\lambda_n - \sum_{\boldsymbol{a}} p(\boldsymbol{a}) \mu_n(\boldsymbol{a})\right)$. Solving zero-sum games with bandit feedback is an active field of research [22, Chapter 9]. The idea is that if the column and row players both use a no-regret algorithm, then the empirical distribution of the action profiles is an $\epsilon_T$-Nash equilibrium (NE) with a vanishing $\epsilon_T$, and the empirical average cost of the row player converges to the value of the game. The convergence rate is then the regret of the row player, divided by $T$. It is an open question whether a convergence rate better than $O\left(T^{-\frac{1}{2}}\right)$ is possible [23, 24], which is achievable if both players use EXP3. To employ this technique in our scenario, one agent will have to operate as a leader that virtualizes the row player that picks $\boldsymbol{p}$ and the column player that picks $n$. This leader agent then broadcasts the chosen action profiles, which are propagated over $G$, so each agent sends $O(N)$ numbers every turn. This approach, however, would fail to provide any QoS regret guarantees. First, the average cost of the row player is not the QoS regret of any of the agents in our game. Second, the fact that the distribution of play $\boldsymbol{p}(T)$ is an $\epsilon_T$-NE does not provide any Euclidean distance convergence rate of $\boldsymbol{p}(T)$ to the exact NE. Hence, there is no bound on the incurred QoS regret of $\boldsymbol{p}(T)$. Remarkably, our queuing approach to solving (1) not only circumvents these issues but achieves, for any $\boldsymbol{\lambda} \in C(\mathcal{G})$, an $\tilde{O}(1)$ empirical average expected QoS regret over $t = 1, \ldots, T$ for all agents, while the expected QoS regret is never worse than $\tilde{O}(\sqrt{t})$ for any $t$ and $n$. Furthermore, our communication overhead is only $O(d(G))$ per agent per turn.

## 3 Dynamic Multiplayer Bandit Algorithm

To tackle the problem in (1), we exploit the similarity between the QoS regret and a queue. We can think of a deterministic arrival flow with rate $\lambda_n$ and a service rate of $r_n(\boldsymbol{a}(t))$ for agent $n$ at turn $t$. Then we can employ ideas from queuing theory to design our algorithm. In particular, max-weight matching of queues to service rates is a centralized algorithm that achieves optimal throughput [25]. Note that in our case the service rate (the reward) is a function of the actions of all agents. Hence, our algorithm has to learn to approximate a max-weight matching distributively for our game model.

---

[1]These numbers are convex combinations of rewards, and their timestamps. They can be quantized with finite resolution without affecting our analysis since we only approximately optimize the weighted sum of rewards anyway. The communication overhead is reported for comparison purposes with other schemes that communicate rewards.

Our algorithm, detailed as Algorithm 1, divides the horizon into epochs. In each epoch, the agents treat their QoS regrets as fixed on their initial value, which is an approximation (Step 1). Based on these values the agents distributedly learn to maximize $\sum_{n=1}^{N} b_n R_n(t) \mu_n(\boldsymbol{a})$ for some positive $\{b_n\}$, using successive elimination (see Lemma 3). Much of the algorithm's efficiency stems from the fact that we do not care what $\{b_n\}$ are, as long as they are positive and constant over epochs. For stochastic time-varying i.i.d. communication graphs (Section 12 in the appendix), $b_n(t)$ will be time-varying as well, and then we need its expectation to be positive and constant over epochs.

Algorithm 1 has a low runtime complexity per turn of $O(\nu_n D_E + S_E)$ for agent $n$, which only computes linear combinations of values from its $\nu_n$ neighbors for $D_E$ turns simultaneously, and goes over a list of $S_E$ action profiles.

---

**Algorithm 1** Dynamic Multiplayer Bandit QoS Learning

---

**Initialization:** Set the epoch index to $e = 0$ and the successive elimination index to $l = 0$. Let $W$ be the communication matrix (Definition 3). Let $S_E = \min\left\{\lceil S_0 \log E \rceil, K^N, E\right\}$, for some $S_0 > 0$, be the random subset size and $D_E > 0$ be the delay parameter. Let $b_{i,j} = \left\{W^{D_E}\right\}_{i,j}$ for all $i, j$.

**Input:** $\lambda_n \geq 0$ and agent index $n$

**For each epoch $e \geq 0$, each agent $n$ runs:**

1. Compute the QoS regret for epoch $e$, $R_n^e = \max\left\{0, \lambda_n t - \sum_{\tau=1}^{t} r_n(\boldsymbol{a}(\tau))\right\}$.

2. Randomize actions $a_n^1, \ldots, a_n^{S_E}$, uniformly and independently over $\{0, \ldots, K-1\}$. Store their indices in the ordered set $\mathcal{B} = \{1, \ldots, S_E\}$, where $\mathcal{B}(l)$ is the $l$-th element.

3. Set $v_i = \hat{\mu}_i = 0$ for all $i = 1, \ldots, S_E$, set $\tilde{r}_n = 1$ and $T_e = eE + 1$.

4. **For $T_e \leq t \leq T_e + E - 1$ ($E$ turns):**

   (a) Set $\theta_n^0 = \tilde{r}_n R_n^e$ and $\theta_m^0 = 0$ for all $m \neq n$.

   (b) Broadcast $\{(\theta_n^\tau, \tau)\}_{\tau=0}^{D_E - 1}$ to neighbors and receive $\{(\theta_m^\tau, \tau)\}_{\tau=0}^{D_E - 1}$ from neighbors.

   (c) **Parallel $D_E$ Consensus Steps:** Set $\theta_n^{\tau+1} = \sum_{m=1}^{N} w_{n,m} \theta_m^\tau, \forall \tau = 0, \ldots, D_E - 1$.

   (d) **Agreement:** If you know $\theta_1^\tau$ for $D_E \leq \tau \leq 2D_E - 1$ then broadcast $(\theta_1^\tau, \tau)$.

   (e) If you receive $(\theta_1^\tau, \tau)$ for $D_E \leq \tau \leq 2D_E - 1$ then set the weighted sum of rewards from $\tau$ turns ago as $S(t - \tau) = \theta_1^\tau$.

   (f) Play $a_n(t) = a_n^{\mathcal{B}(l)}$ and set $\tilde{r}_n = r_n(\boldsymbol{a}(t))$.

   (g) Update the number of visits $v_k \leftarrow v_k + 1$ to $\boldsymbol{a}^k = \left(a_1^k, \ldots, a_N^k\right)$ for $k = \mathcal{B}(l - 2D_E)$.

   (h) Compute the confidence radius $\rho_k = \sqrt{\frac{2\left(\sum_{n=1}^{N} b_{1,n} R_n^e\right)^2 \log E}{\max\{v_k, 1\}}}$ and

   $$\hat{\mu}_k \leftarrow \left(1 - \frac{1}{v_k}\right)\hat{\mu}_k + \frac{S(t - 2D_E)}{v_k} \quad , \quad L_k = \hat{\mu}_k - \rho_k \quad , \quad U_k = \hat{\mu}_k + \rho_k.$$

   (i) **Elimination:** If $l = |\mathcal{B}|$ then delete from $\mathcal{B}$ all $i$ such that $U_i < \max_{j \in \mathcal{B}} L_j$, and set $l = 0$.

   (j) Update the successive elimination index $l \leftarrow l + 1$.

5. Update the epoch index $e \leftarrow e + 1$.

**End**

---

### 3.1 Communication

To know the reward values of others, agents need to communicate. We assume that agents can send numbers over a communication graph $G$, weighted by a row-stochastic communication matrix $W$:

**Definition 3.** Let $G$ be the adjacency matrix of an undirected connected graph, such that $G_{n,m} = 1$ if agent $n$ and agent $m$ can communicate or that $n = m$. Let $\nu_n$ be the degree of agent $n$ in $G$. Let $W$ be a matrix such that $w_{n,m} > 0$ if and only if $G_{n,m} = 1$ and $\sum_{m=1}^{N} w_{n,m} = 1$ for all $n$.

A key design question is how much should agents know about each other to achieve (1). A max-weight matching at turn $t$ is an $\boldsymbol{a}$ that maximizes $\sum_{n=1}^{N} b_n R_n(t) \mu_n(\boldsymbol{a})$ for some positive $\{b_n\}$. A

naive choice is to guarantee that all agents know all rewards before taking an action. This would require each agent to send $O(N)$ numbers per turn, so it is infeasible in large-scale networks. Instead, our algorithm requires a communication overhead of $O(d(G))$, without sacrificing performance. This dramatic save in communication is possible since to guarantee stability, agents do not need to have the same coefficient $b_n$ in the sum. Hence, in Algorithm 1, agents run the consensus only until all local averages include all $\{R_n^e r_n(\boldsymbol{a}(t))\}_n$ (where $R_n^e \approx R_n(t)$), even if not with equal $b_n$.

In contrast to the distributed optimization literature [26], we do not need the communication matrix $W$ to be doubly stochastic, which is hard to guarantee distributedly [27]. Since $W$ is not doubly stochastic, our consensus cannot converge to the average reward. In fact, since we stop the consensus early after exactly $D_E$ turns, it does not converge at all. However, after our consensus step, each agent has a weighted sum of rewards that includes all agents that have a positive QoS, with weights that depend on $W$. In the next $D_E$ turns, agents propagate (Steps 4(d) and 4(e)) the weighted sum of rewards that Agent 1 obtained, such that they all eventually know it with zero error.

Our algorithm introduces delays so that agents can agree on $\sum_{n=1}^N b_{1,n} R_n^e r_n(\boldsymbol{a}(t+1))$, that Agent 1 obtained after the distributed averaging step (Steps 4(b) and 4(c)). If the diameter of $G$ is $d(G)$, then a minimum of $d(G)$ turns is necessary to guarantee that all agents contributed to $\sum_{n=1}^N b_{1,n} R_n^e r_n(\boldsymbol{a}(t+1))$. However, it is unnecessary to continue the consensus after this point, which is a major advantage of our consensus scheme over the classical one.

In Steps 4(b)-4(e), agents pipeline $D_E$ simultaneous consensus algorithms that cannot be mixed since they are on the rewards generated at different turns from different action profiles.

In a connected graph $d(G) \leq N$ so $D_E = N$ is always sufficient. However, usually, a tighter bound on $d(G)$ is available. For a random Erdős–Rényi graph $G(N,p)$, $d(G) = O\left(\frac{\log N}{\log(Np)}\right)$ with high probability [28], and $G(N,p)$ is connected with high probability for $p = \frac{c \log N}{N}$ with $c > 1$.

In our scenario, an apriori bound on $\sum_{n=1}^N b_{1,n} R_n^e r_n(\boldsymbol{a})$, the effective reward for the successive elimination, is unknown to the agents but is needed to adjust the confidence radius $\rho_k$. Hence, at the beginning of each epoch (Step 4(a)), agents start a consensus averaging on $\{R_n^e\}_n$ to obtain $\sum_{n=1}^N b_{1,n} R_n^e$, which bounds $\sum_{n=1}^N b_{1,n} R_n^e r_n(\boldsymbol{a})$, but varies between epochs.

### 3.2   Distributed and Scalable Successive Elimination

In the second part of every epoch (Steps 4(f)-4(j)), agents use the values they got in the communication step to run distributed successive elimination on a random subset of the action profiles. In our case, the effective reward of "arm" $\boldsymbol{a}$ at epoch $e$ is $\sum_{n=1}^N b_{1,n} R_n^e r_n(\boldsymbol{a})$. In Step 2, each agent prepares a sequence of $S_E$ actions independently and uniformly at random. This induces a sequence of $S_E$ action profiles that all agents agree on their indices $\mathcal{B} = \{1, \ldots, S_E\}$ (which can include duplicates). Therefore, all agents count the same number of visits $v_{\boldsymbol{a}}(t)$ for all $\boldsymbol{a}$ in Step 4(g). Moreover, all agents agree on the same weighted sum of rewards for the action profile they played $2D_E$ turns ago, denoted by $\hat{\mu}_k$ for $k = \mathcal{B}(l - 2D_E)$. Hence, all agents compute the same confidence radius $\rho_k$ in Step 4(h), for all the action profiles indexed in $\mathcal{B}$. As a result, all agents agree on when to eliminate an action profile from $\mathcal{B}$ in Step 4(i) and will skip it in their next cycle over the action profiles, which is indexed by $l$.

Restricting the search to a random subset of action profiles is done to improve the scalability of the algorithm, as we discuss in detail in Section 7 of the appendix. For $S_E = K^N$, no randomization is needed as agents can go over all the action profiles sequentially.

### 3.3   QoS Regret Guarantees

Theorem 2 below bounds the expected QoS regrets of the agents. The proof is given in the appendix. If the epoch length $E_T$ increases with $T$, then the theorem shows that $\mathbb{E}\{R_n(t)\} = O\left(\sqrt{E_T t}\right)$ for any $n, t$ and any $\boldsymbol{\lambda} \in C(\mathcal{G})$. However, this does not provide a guarantee on the fraction of time during which the QoS regret of a specific agent is "large". A highly fluctuating QoS regret means that an agent has to wait a long time to be "compensated" for when it was patient in favor of others. Hence, we also provide a general bound on $\frac{1}{T} \sum_{t=0}^{T-1} \sum_{n=1}^N b_{1,n} \mathbb{E}\{R_n(t)\}$, for positive $\{b_{1,n}\}$ that are specified below. Using this bound, the first part of Theorem 2 assumes a fixed epoch length $E$

and that $\boldsymbol{\lambda} \in C_{\varepsilon_E}(\mathcal{G})$ for $\varepsilon_E$ in (5), so $\boldsymbol{\lambda}$ is far enough from the boundary of $C(\mathcal{G})$. In this case, we obtain $\limsup_{T \to \infty} \frac{1}{T} \sum_{t=0}^{T-1} \sum_{n=1}^{N} \mathbb{E}\{R_n(t)\} < \infty$, so the empirical average of the expected QoS regret of any agent is $O(1)$. In the queuing literature, these stability notions are known as mean rate stability (i.e., sublinear expected QoS regret) and strong stability.

**Theorem 2.** *Let $\mathcal{G}$ be a multiplayer multi-armed bandit game, played for $T$ turns. Let $\boldsymbol{\lambda} \in C(\mathcal{G})$. Let the agents run Algorithm 1 with a row-stochastic $W$, a connected $G$ with diameter $d(G)$, epochs of length $E$, a random subset of action profiles of size $S_E \leq \lceil S_0 \log E \rceil$ for some $S_0 > 0$ and $D_E \geq d(G)$. Let $b_{i,j} = \left\{W^{D_E}\right\}_{i,j}$. Define the set of approximately optimal action profiles*

$$\mathcal{A}^*(\boldsymbol{\alpha}) = \left\{ \boldsymbol{a} \mid \sum_{n=1}^{N} \alpha_n \mu_n(\boldsymbol{a}) \geq \max_{\boldsymbol{a}'} \sum_{n=1}^{N} \alpha_n \mu_n(\boldsymbol{a}') - \frac{\log E}{\sqrt{E}} \right\} \tag{4}$$

*and*

$$\varepsilon_E = \frac{33(S_0 + 1)\log E}{\sqrt{E}} + 2\frac{D_E}{E} + E^{-\frac{S_0 \min_{\boldsymbol{\alpha} \in \Delta^N} |\mathcal{A}^*(\boldsymbol{\alpha})|}{K^N}}. \tag{5}$$

*Define*

$$\delta(\boldsymbol{\lambda}, E) = \max_{\boldsymbol{p} \in \Delta^{K^N}} \min_{n} \left( \sum_{\boldsymbol{a}} p(\boldsymbol{a}) \mu_n(\boldsymbol{a}) - \lambda_n - \varepsilon_E \right). \tag{6}$$

*If $\boldsymbol{\lambda} \in C_{\varepsilon_E}(\mathcal{G})$ then the QoS regrets (Definition 2) satisfy*

$$\frac{1}{T} \sum_{t=0}^{T-1} \sum_{n=1}^{N} b_{1,n} \mathbb{E}\{R_n(t)\} \leq \frac{1 + (2 + \varepsilon_E)E}{\delta(\boldsymbol{\lambda}, E)} \tag{7}$$

*where $\mathbb{E}\{\cdot\}$ is over the rewards and the random subsets, and $b_{1,n} > 0$ for all $n$.*

*Therefore, for any $\boldsymbol{\lambda} \in C_{\varepsilon_E}(\mathcal{G})$, Algorithm 1 achieves $O(1)$ empirical average expected QoS regret for all $n$ ,i.e., $\limsup_{T \to \infty} \frac{1}{T} \sum_{t=0}^{T-1} \sum_{n=1}^{N} \mathbb{E}\{R_n(t)\} < \infty$ (strong stability) and $\mathbb{E}\{R_n(t)\} = O(\sqrt{t})$ for all $n, t$. Furthermore, if we pick $E_T$ such that $E_T \to \infty$ and $\frac{E_T}{T} \to 0$ as $T \to \infty$, then for any $\boldsymbol{\lambda} \in C(\mathcal{G})$, Algorithm 1 achieves $\frac{1}{T} \sum_{t=0}^{T-1} \sum_{n=1}^{N} \mathbb{E}\{R_n(t)\} = O(E_T)$ for all $n$ and $\mathbb{E}\{R_n(t)\} = O(\sqrt{E_T t})$ for all $n, t$ (mean rate stability).*

## 4 QoS Regret Stability Analysis

In this section, we break our analysis required to prove Theorem 2 into lemmas and postpone the proofs to the appendix. The proof technique is inspired by the optimality proof of max-weight matching [25]. However, the QoS regret $R_n(t)$ does not behave like a standard queue. The main difference is that the service rate of agent $n$, which is its reward function, is not selected by a centralized decision-maker but is a function of the actions of the distributed agents. As opposed to a standard queue, the "backlog" of the QoS regret is not a sufficient statistic, since one has to remember $\lambda_n t - \sum_{\tau=1}^{t} r_n(\tau)$ even if $R_n(t) = 0$. Since $\lambda_n t - \sum_{\tau=1}^{t} r_n(\tau)$ can get very negative "empty queues" ($R_n(t) = 0$) are much more likely in our problem. As we discuss in Section 7 in the appendix, this fact can be used to improve the scalability of our algorithm, since agents with $R_n(t) = 0$ can be thought of as "passive". Moreover, our "arrival process" is deterministic, of $\lambda_n < 1$ reward points every turn for agent $n$, and the backlog is not an integer. As detailed next, some of these differences require careful analysis or affect the algorithm design. For example, while a sublinearly increasing queue (in $T$) is not strongly stable, in our case, as long as $\frac{R_n(T)}{T} \to 0$, agent $n$ will asymptotically achieve its desired QoS in the empirical average sense.

**Definition 4.** Define the matrix $B = W^{D_E}$, and denote its $(i, j)$ element by $b_{i,j}$. Define the following Lyapunov function such that $L(0) = 0$ and for all $t \geq 1$

$$L(t) \triangleq \sum_{n=1}^{N} b_{1,n} R_n^2(t). \tag{8}$$

$L(t)$ only uses the coefficients of Agent 1 since all agents, that know their IDs, agree on the weighted sum of rewards that Agent 1 sees. This forced agreement ensures that all agents will use the exact same weighted sum of rewards. However, Agent 1 does not know more than others and can be replaced by any of them.

The first lemma bounds the Lyapunov drift in each turn.

**Lemma 1.** *The Lyapunov drift at turn $t \geq 0$ satisfies:*

$$L(t+1) - L(t) \leq 1 + 2\sum_{n=1}^{N} b_{1,n} R_n(t) (\lambda_n - r_n(\boldsymbol{a}(t+1))). \tag{9}$$

The proof of Theorem 2 follows by upper bounding $\mathbb{E}\{L(T)\} - \mathbb{E}\{L(0)\}$, where a smaller bound implies smaller QoS regrets. The expectation (over the reward) of the upper bound of Lemma 1 is minimized by the maximum weight matching $\boldsymbol{a}^*(t+1) = \arg\max_{\boldsymbol{a}} \sum_{n=1}^{N} b_{1,n} R_n(t) \mu_n(\boldsymbol{a})$. The two main obstacles our algorithm has to overcome to approximate max-weight matching are the distributed nature of the agents, and the unknown reward functions (bandit feedback). Distributed agents cannot deduce $\boldsymbol{a}^*(t+1)$ instantly every turn since they need time to agree on the weighted sum of rewards. For this reason, our algorithm works in epochs. Working in epochs means that our algorithm treats the QoS regret $R_n(t)$ as constant on its initial value $R_n^e \triangleq R_n(T_e - 1)$ throughout epoch $e$. The agents can then aim to solve the following problem instead in every epoch

$$\boldsymbol{a}_e^* = \arg\max_{\boldsymbol{a}} \sum_{n=1}^{N} b_{1,n} R_n^e \mu_n(\boldsymbol{a}). \tag{10}$$

The next lemma bounds the resulting error from solving (10) instead of the max-weight matching.

**Lemma 2.** *Define $e(t)$ as the epoch such that $T_{e(t)} \leq t \leq T_{e(t)} + E - 1$. Then*

$$\sum_{t=0}^{T-1} \sum_{n=1}^{N} b_{1,n} \mu_n\left(\boldsymbol{a}_{e(t)}^*\right) R_n^{e(t)} \geq \sum_{t=0}^{T-1} \sum_{n=1}^{N} b_{1,n} \mu_n(\boldsymbol{a}^*(t+1)) R_n(t) - ET\max_n \lambda_n. \tag{11}$$

In addition to solving (10) instead of the max-weight matching, working in epochs incurs an additive error of $O(ET)$ again while replacing $R_n(t)$ with $R_n^e$ while bounding the drift in (9).

The other loss of Algorithm 1 stems from learning $\boldsymbol{a}_e^*$ with bandit feedback. While dividing into epochs creates an additive error of $O(ET)$, learning the optimal action profile $\boldsymbol{a}_e^*$ creates a multiplicative error, so the larger the QoS regrets $R_n^e$ are, the larger this error is.

For this reason, we need to employ an algorithm that achieves sublinear regret in learning $\boldsymbol{a}_e^*$. Algorithm 1 induces successive elimination on a random subset of $S_E$ action profiles, with $\sum_{n=1}^{N} b_{1,n} R_n^e r_n(\boldsymbol{a})$ as the reward function. The next lemma bounds the error (regret) of this successive elimination. Since agents only agree on $\sum_{n=1}^{N} b_{1,n} R_n^e r_n(\boldsymbol{a}(t))$ after $2D_E$ turns, our successive elimination suffers from delayed feedback. While not our focus, we chose successive elimination over UCB since it is more robust to delays, as was shown in [29]. The restriction to a random subset of $S_E$ action profiles is done to improve scalability, but it leads to a worse regret than in the single-agent standard multi-armed bandits. Nevertheless, the regret is still sublinear in $E$ and therefore good enough to prove our QoS regret bounds.

**Lemma 3.** *Let $R_n^e = R_n(T_e - 1)$ and define the set of approximately optimal action profiles as*

$$\mathcal{A}_e^* = \left\{ \boldsymbol{a} \mid \frac{\sum_{n=1}^{N} b_{1,n} R_n^e \mu_n(\boldsymbol{a})}{\sum_{n=1}^{N} b_{1,n} R_n^e} \geq \max_{\boldsymbol{a}'} \frac{\sum_{n=1}^{N} b_{1,n} R_n^e \mu_n(\boldsymbol{a}')}{\sum_{n=1}^{N} b_{1,n} R_n^e} - \frac{\log E}{\sqrt{E}} \right\}. \tag{12}$$

*Let $\boldsymbol{a}_e^* \in \mathcal{A}_e^*$. Define the filtration $\mathcal{F}_e = \sigma\left(\left\{\boldsymbol{a}(t), \{z_n(t)\}_{n\in\mathcal{N}} \mid t < T_e\right\}\right)$. Then Algorithm 1 maintains*

$$\mathbb{E}\left\{\sum_{t=T_e}^{T_e+E-1} \sum_{n=1}^{N} b_{1,n} R_n^e \left(\mu_n(\boldsymbol{a}_e^*) - r_n(\boldsymbol{a}(t+1))\right) \mid \mathcal{F}_e\right\} \leq$$

$$\left[33(S_0+1)\log E\sqrt{E} + 2D_E + E^{1 - \frac{S_0|\mathcal{A}_e^*|}{KN}}\right] \sum_{n=1}^{N} b_{1,n} R_n^e \tag{13}$$

*where $\mathbb{E}\{\cdot \mid \mathcal{F}_e\}$ is over the rewards and the $e$-th subset of action profiles.*

# 5 Numerical Simulations

Our algorithm is useful in any case where cooperative agents need to achieve a target QoS over time. Next, we tested our algorithm for wireless channel access and multitasking robots, two applications where smart time-sharing can significantly improve performance. For each experiment, we plot the sum of QoS regrets averaged over 100 realizations with one standard deviation shaded region.

## 5.1 Wireless Channel Access

Consider $N$ devices and $K - 1$ channels, such that each device can choose to transmit a packet on channel $k \geq 1$ by playing $a_n = k$ or not to transmit at all by playing $a_n = 0$. We consider a "soft collisions" model where the success probability of transmitting a packet for device $n$ on channel $k$ is $p_n^k = \frac{g_{n,n}^k}{1 + \sum_{m=1}^N g_{m,n} 1\{a_m = k\}}$ for some channel gains $\{g_{n,m}\}$ and $\{g_{n,n}^k\}$. This model generalizes the collision model [12, 14, 15], where multiple devices that pick the same channel all receive zero reward. As in practice, the geometry of the devices dictates how much they interfere with each other.

Fig. 1(a) shows the sum of QoS regrets over time for $N = 4$ and $K = 3$. We used $S_E = 128$, $d(G) = 3$ and $E = 50 S_E$. The channel gains were $g_{1,4} = g_{1,3} = g_{2,4} = g_{3,2} = 0$ and $g_{1,2} = g_{3,4} = 100$ such that $g_{n,m} = g_{m,n}$ for all $n, m$, and $g_{1,1}^2 = g_{2,2}^2 = g_{3,3}^1 = g_{4,4}^1 = 0.1$, $g_{1,1}^1 = g_{2,2}^1 = g_{3,3}^2 = g_{4,4}^2 = 1$. Hence devices 1 and 2 are in proximity, and devices 3 and 4 are also in proximity, but far away from 1 and 2. Then, devices 1 and 2 prefer channel $k = 1$ while devices 3 and 4 prefer channel $k = 2$. The QoS vector was $\lambda_n = 0.499$ for all $n$, which cannot be achieved with a static solution, but only if devices 1 and 3 transmit simultaneously each on their preferred channel, and in the next turn devices 2 and 4 do the same, and so on in a round-robin manner. After a while, the QoS regrets stabilized on low values.

Fig. 1(b) shows the sum of QoS regrets over time for $N = 100$ and $K = 2$ (one channel). The channel gains $\{g_{n,m}\}$ and $\{g_{n,n}^k\}$ were chosen independently and uniformly at random on $[0, 1]$. We used $S_E = 10^4$, $E = 5 S_E$, and $d(G) = 4$. The QoS of agent $n$ was $\lambda_n = 1.9 \frac{p_n^1 - 0.02}{N}$, which is 90% more than what an almost perfect time-sharing can achieve. The stability of the QoS regrets is again evident, in agreement with the theory. Note that while there are $K^N = 2^{100}$ action profiles, considering only a random subset of $S_E = 10^4$ action profiles was enough to achieve this QoS.

## 5.2 Multitasking Robots

Consider $N$ robots and $K = N$ tasks, such that the $n$-th robot is in charge of the $n$-th task. Each robot $n$ has a skill of $s_n^k$ for the $k$-th task. Each turn, each robot can choose to work on the $k$-th task by playing $a_n = k$. The reward of robot $n$ is $r_n(\boldsymbol{a}) = \sum_{m=1}^N 1_{\{a_m = n\}} s_m^n + z_n$, which models the rate at which this task progresses, where $z_n$ is a uniform noise on $[-0.2, 0.2]$ that is i.i.d. over time and independent between agents. Hence, only one robot can measure the progress of a certain task, but the others can still assist.

Fig. 2(a) shows the sum of QoS regrets over time for $N = K = 4$. We used $E = 5K^N = 5120$, $d(G) = 3$. The QoS vector was $\boldsymbol{\lambda} = (0.8, 0.2, 0.4, 0.4)$ and the skill matrix was $\begin{pmatrix} 0.5 & 0.4 & 0.2 & 0.2 \\ 0.5 & 0.4 & 0.2 & 0.3 \\ 0.25 & 0.5 & 0.6 & 0.3 \\ 0.25 & 0.4 & 0.2 & 0.6 \end{pmatrix}$ (e.g., $s_1^4 = 0.2$). As a result, $\lambda_1$ is larger than the maximal skill of a single robot so the robots had to cooperate on their tasks to achieve this QoS vector. Additionally, no static solution can support this $\boldsymbol{\lambda}$.

Fig. 2(b) shows the sum of QoS regrets for $N = 8$ and $K = 8$. We used $S_E = 1000$, $d(G) = 7$ and $E = 10 S_E$, $\boldsymbol{\lambda} = (0.83, 0.79, 0.78, 0.71, 0.85, 0.76, 0.74, 0.86)$ and picked $s_n^k$ uniformly and independently at random on $[0.2, 1]$. Note that while there are $K^N = 8^8$ action profiles, considering only a random subset of $S_E = 10^3$ action profiles was enough to achieve this QoS.

In both figures, the sum of QoS regrets increases slowly with $T$, which for a traditional queue would mean no strong stability but in our case means that the empirical average rewards converge to $\boldsymbol{\lambda}$.

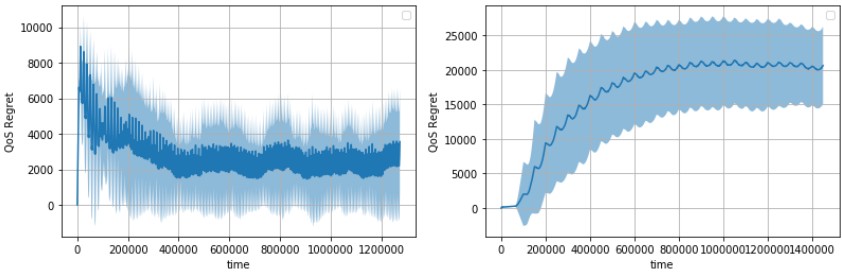

Figure 1: Sum of QoS regrets: channel access, $N = 4, K = 3$ (left) and $N = 100, K = 2$ (right)

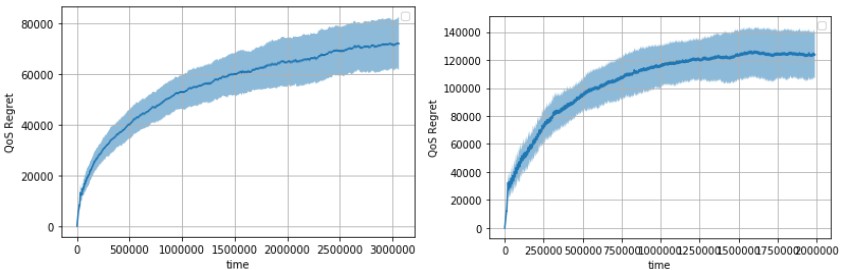

Figure 2: Sum of QoS regrets: multitasking robots, $N = 4$ (left) and $N = 8$ (right)

## 6 Conclusion

We studied how a team of agents can improve the Quality of Service (QoS) of each by inducing a dynamic behavior between them. The main observation that inspired our algorithm is that the QoS regret of each agent behaves like a queue. Then, our distributed algorithm learns to approximate a max-weight matching of agents to actions to maximize the "throughout" going out of these virtual queues. To that end, our algorithm works in epochs and employs simultaneous consensus averaging and distributed successive elimination on a random subset of action profiles and with delayed feedback. Our main result shows that our algorithm can achieve all QoS vectors in the capacity region for large enough $T$, where all agents have an empirical average expected QoS regret of $\tilde{O}(1)$ over time which never exceeds $\tilde{O}(\sqrt{t})$ for any $t$. Simulations demonstrate the effectiveness of our algorithm for channel access in wireless networks and multitasking robots.

We then extended our result for stochastic communication graphs that are i.i.d. over time. The communication graphs create dependencies between the rewards of the "arms" (i.e., action profiles) over time. While we were able to bypass these dependencies for the i.i.d. case, a Markov graph process, for example, will lead to a restless bandits scenario instead where successive elimination does not work. While in principle it is easy to replace the successive elimination part of our algorithm, the new bandit algorithm will have to be distributed, ideally with a small communication overhead.

We considered an arbitrary discrete game where an exhaustive search is necessary to find an approximately optimal action profile. However, applications often focus on more structured games. For example, in congestion games, best-response dynamics are guaranteed to converge to a Nash equilibrium. Hence, if a congestion game is designed such that all Nash equilibria are efficient, an exhaustive search is not needed [30]. Alternatively, the agents can directly solve their combinatorial optimization problem as a team, leveraging its structure. Examples are the assignment problem, unconstrained submodular minimization, and shortest path [31]. Then the challenge is to design a distributed algorithm to solve the combinatorial optimization based only on bandit feedback.

Our work introduces an analogy between multiagent QoS and queuing. By further studying this analogy we can leverage more tools from queuing theory to design dynamic distributed learning algorithms with strong performance guarantees. For example, scheduling algorithms with delay guarantees [32] can help to design QoS multiagent algorithms with better delay performance. Another example is to employ the "drift plus penalty" technique [25] to optimize an objective such as the sum of rewards while still guaranteeing some QoS for all agents.

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
