# 7 Scalability in the Number of Players and Actions

As opposed to a single agent with $K$ arms, the number of action profiles is $K^N$ which explodes for a large-scale network. Successive elimination (or UCB), even with no noise, requires the agents to go over all $K^N$ possible action profiles. This search does not affect the computational complexity, which is $O\left(\nu_n D_E + S_E\right)$ for agent $n$ that computes $D_E$ parallel consensus steps and goes over a list of $S_E$ action profiles. However, this exhaustive search does affect the scalability of the algorithm since it factors the QoS regret. Intuitively, we would need $E \geq K^N$ to find the optimal action profile even with no noise, which creates delays where agents have to wait for their average reward to go above their $\lambda_n$. Next, we discuss two mechanisms that improve the scalability of our algorithm.

## 7.1 Stability yields scalability

Fortunately, the dynamic nature of our problem inherently improves scalability. During a normal operation of a stable system, only a small portion of agents have $R_n^e > 0$. All other agents with $R_n^e = 0$ do not care about their reward during epoch $e$. This fact dramatically shrinks the set of action profiles that can be optimal. For example, when $a_n$ represents the "demand" of agent $n$, agents with $R_n^e = 0$ will trivially play $a_n = 0$ in the action profile that maximizes $\sum_{n=1}^{N} b_n R_n^e \mu_n\left(\boldsymbol{a}\right)$. Hence, the mean performance of the algorithm is considerably better when the QoS vector is deep inside the interior of the capacity region. This is often the case with the advertised QoS in applications since the infrastructure, and therefore the capacity, is typically designed for peak demand.

This is also the case in our numerical simulations of Section 5, which we used in our implementation. In the channel access game, if agent $n$ has $R_n^e = 0$, then the optimal action profile $\boldsymbol{a}_e^*$ for this epoch has to satisfy $a_{e,n}^* = 0$. If $\boldsymbol{\lambda}$ is a safe margin away from the boundary of the capacity region $\mathcal{C}\left(\mathcal{G}\right)$, then most agents will have $R_n^e = 0$ most of the time. Therefore, the effective number of action profiles is reduced to $K^{|\{n|R_n^e > 0\}|}$. In the multitasking robots game, if agent $n$ has $R_n^e = 0$, then the optimal action profile $\boldsymbol{a}_e^*$ has to satisfy $a_{e,m}^* \neq n$ for all $m$. If $\boldsymbol{\lambda}$ is a safe margin away from the boundary of $C\left(\mathcal{G}\right)$, then most agents will have $R_n^e = 0$ most of the time. Therefore, the effective number of action profiles is reduced to $|\{n \mid R_n^e > 0\}|^N$, given that the agents in $\{n \mid R_n^e > 0\}$ broadcast their indices at the beginning of the epoch.

## 7.2 Restricting the search to a limited random subset of action profiles

Inspired by [33], to improve scalability, our agents confine their search to a random subset $\mathcal{S}_E$ of action profiles they distributedly pick. Hence, their performance depends on the best action profile in $\mathcal{S}_E$. In Section 4, we bound the probability that $\mathcal{S}_E$ does not include an $\Delta_E$-optimal action profile for some $\Delta_E > 0$ (not needed by the algorithm). For this reason, our $\varepsilon_E$ bound in (5) uses $m^* \triangleq \min_{\boldsymbol{\alpha} \in \Delta^N} |\mathcal{A}^*\left(\boldsymbol{\alpha}\right)|$, which is the minimal number of $\Delta_E$-optimal solutions of $\sum_{n=1}^{N} \alpha_n \mu_n\left(\boldsymbol{a}\right)$, over all $\boldsymbol{\alpha}$ in the $N$-dimensional simplex $\Delta^N$. This $m^*$ is a worst-case bound for $|\mathcal{A}_e^*|$ in (12), and statistically most simplex vectors $\boldsymbol{\alpha} \in \Delta^N$ are not common. In large-scale systems, $|\mathcal{A}_e^*|$ is typically large. If most QoS regrets are zero then the effective dimension of the problem shrinks, making $|\mathcal{A}_e^*|$ large. If on the other hand almost all QoS regrets are non-zero, then $\alpha_n = O\left(\frac{1}{N}\right)$ for most $n$ and $\sum_{n=1}^{N} \alpha_n \mu_n\left(\boldsymbol{a}\right)$ is robust to changes in $\mu_n$ for a few values of $n$, so $|\mathcal{A}_e^*|$ is again large. Furthermore, symmetry in the problem leads to large $m^*$. As a simple example, with $N$ identical resources as the actions all permutations are optimal and $m^* \geq N!$.

For finite $T$, we can choose $E$ to optimize the tradeoff between $\varepsilon_E$ (capacity region) and $\left(2 + \varepsilon_E\right) E$ (delay). A large $E$ makes $\varepsilon_E$ small and therefore enlarges the approximated capacity region $C_{\varepsilon_E}\left(\mathcal{G}\right)$, but increases the expected QoS regret, which amounts to the time an agent has to wait to get its "fair share". A small $E$ minimizes this delay, but shrinks $C_{\varepsilon_E}\left(\mathcal{G}\right)$. Given some statistical knowledge about the game, we can minimize $\varepsilon_E$ which improves both capacity and delay. This can be done by estimating a lower bound on $m^*$ and using it to pick the smallest $S_0$ such that $\varepsilon_E$ is small enough (see (5)). This does not require knowing the reward functions, since large $N$ can give concentration results on the sum of weighted rewards. An example is knowing the distribution of the random locations of the agents without knowing the locations themselves. Nevertheless, our main result in Theorem 2 does not require any knowledge of the game and holds for all $E$ (that can be a function of $T$).

# 8   Proof of Lemma 1

By definition

$$L\left(t+1\right) - L\left(t\right) =$$

$$\sum_{n=1}^{N} b_{1,n} \max\left\{0, \left(\lambda_n t - \sum_{\tau=1}^{t} r_n\left(\tau\right) + \lambda_n - r_n\left(t+1\right)\right)\right\}^2$$

$$- \sum_{n=1}^{N} b_{1,n} \max\left\{0, \left(\lambda_n t - \sum_{\tau=1}^{t} r_n\left(\tau\right)\right)\right\}^2 \triangleq \sum_{n=1}^{N} b_{1,n} B_n. \quad (14)$$

If $\lambda_n t - \sum_{\tau=1}^{t} r_n\left(\tau\right) + \lambda_n - r_n\left(t+1\right) > 0$ and $\lambda_n t - \sum_{\tau=1}^{t} r_n\left(\tau\right) > 0$ then

$$B_n = \left(\lambda_n - r_n\left(t+1\right)\right)^2 + 2\left(\lambda_n - r_n\left(t+1\right)\right)\left(\lambda_n t - \sum_{\tau=1}^{t} r_n\left(\tau\right)\right)$$

$$+ \left(\lambda_n t - \sum_{\tau=1}^{t} r_n\left(\tau\right)\right)^2 - \left(\lambda_n t - \sum_{\tau=1}^{t} r_n\left(\tau\right)\right)^2 =$$

$$\left(\lambda_n - r_n\left(t+1\right)\right)^2 + 2\left(\lambda_n - r_n\left(t+1\right)\right)\left(\lambda_n t - \sum_{\tau=1}^{t} r_n\left(\tau\right)\right)^+. \quad (15)$$

If $\lambda_n t - \sum_{\tau=1}^{t} r_n\left(\tau\right) + \lambda_n - r_n\left(t+1\right) \le 0$ and $\lambda_n t - \sum_{\tau=1}^{t} r_n\left(\tau\right) \le 0$ then

$$B_n = 0 \le \left(\lambda_n - r_n\left(t+1\right)\right)^2 = \left(\lambda_n - r_n\left(t+1\right)\right)^2 +$$

$$2\left(\lambda_n - r_n\left(t+1\right)\right)\left(\lambda_n t - \sum_{\tau=1}^{t} r_n\left(\tau\right)\right)^+ \quad (16)$$

since $\left(\lambda_n t - \sum_{\tau=1}^{t} r_n\left(\tau\right)\right)^+ = 0$ in this case.

If $\lambda_n t - \sum_{\tau=1}^{t} r_n\left(\tau\right) + \lambda_n - r_n\left(t+1\right) > 0$ and $\lambda_n t - \sum_{\tau=1}^{t} r_n\left(\tau\right) \le 0$ then

$$0 \le \lambda_n t - \sum_{\tau=1}^{t} r_n\left(\tau\right) + \lambda_n - r_n\left(t+1\right) \le \lambda_n - r_n\left(t+1\right)$$

$$\implies B_n \le \left(\lambda_n - r_n\left(t+1\right)\right)^2 =$$

$$\left(\lambda_n - r_n\left(t+1\right)\right)^2 + 2\left(\lambda_n - r_n\left(t+1\right)\right)\left(\lambda_n t - \sum_{\tau=1}^{t} r_n\left(\tau\right)\right)^+ \quad (17)$$

since $\left(\lambda_n t - \sum_{\tau=1}^{t} r_n\left(\tau\right)\right)^+ = 0$ in this case.

If $\lambda_n t - \sum_{\tau=1}^{t} r_n\left(\tau\right) + \lambda_n - r_n\left(t+1\right) \le 0$ and $\lambda_n t - \sum_{\tau=1}^{t} r_n\left(\tau\right) > 0$ then

$$0 \le \left(\lambda_n t + \lambda_n - \sum_{\tau=1}^{t} r_n\left(\tau\right) - r_n\left(t+1\right)\right)^2 =$$

$$\left(\lambda_n t - \sum_{\tau=1}^{t} r_n\left(\tau\right)\right)^2 + \left(\lambda_n - r_n\left(t+1\right)\right)^2 + 2\left(\lambda_n - r_n\left(t+1\right)\right)\left(\lambda_n t - \sum_{\tau=1}^{t} r_n\left(\tau\right)\right)^+ \quad (18)$$

and so

$$B_n = -\left(\lambda_n t - \sum_{\tau=1}^{t} r_n\left(\tau\right)\right)^2 \leq$$

$$\left(\lambda_n - r_n\left(t+1\right)\right)^2 + 2\left(\lambda_n - r_n\left(t+1\right)\right)\left(\lambda_n t - \sum_{\tau=1}^{t} r_n\left(\tau\right)\right)^{+}. \quad (19)$$

We conclude that in any case, for all $n$,

$$B_n \leq \left(\lambda_n - r_n\left(t+1\right)\right)^2 + 2\left(\lambda_n - r_n\left(t+1\right)\right)\left(\lambda_n t - \sum_{\tau=1}^{t} r_n\left(\tau\right)\right)^{+}. \quad (20)$$

Hence we can bound the drift in the Lyapunov function as follows

$$L\left(t+1\right) - L\left(t\right) \leq$$

$$\sum_{n=1}^{N} b_{1,n}\left(\lambda_n - r_n\left(t+1\right)\right)^2 + 2\sum_{n=1}^{N} b_{1,n}\left(\lambda_n - r_n\left(t+1\right)\right)\left(\lambda_n t - \sum_{\tau=1}^{t} r_n\left(\tau\right)\right)^{+} \underset{(a)}{\leq}$$

$$1 + 2\sum_{n=1}^{N} b_{1,n}\lambda_n\left(\lambda_n t - \sum_{\tau=1}^{t} r_n\left(\tau\right)\right)^{+} - 2\sum_{n=1}^{N} b_{1,n}r_n\left(t+1\right)\left(\lambda_n t - \sum_{\tau=1}^{t} r_n\left(\tau\right)\right)^{+} =$$

$$1 + 2\sum_{n=1}^{N} b_{1,n}\lambda_n R_n\left(t\right) - 2\sum_{n=1}^{N} b_{1,n}r_n\left(t+1\right)R_n\left(t\right) \quad (21)$$

where (a) uses that $0 \leq r_n\left(\boldsymbol{a}\right) \leq 1$, $0 \leq \lambda_n \leq 1$ and $\sum_{n=1}^{N} b_{1,n} = 1$.

## 9  Proof of Lemma 2

Let $\lambda_{\max} = \max_{n}\lambda_n$. For all $T_e \leq t \leq T_e + E - 1$ we have

$$\max_{\boldsymbol{a}} \sum_{n=1}^{N} b_{1,n}\mu_n\left(\boldsymbol{a}\right)R_n^e = \max_{\boldsymbol{a}}\left[\sum_{n=1}^{N} b_{1,n}\mu_n\left(\boldsymbol{a}\right)\left(R_n\left(t\right) + R_n^e - R_n\left(t\right)\right)\right] \geq$$

$$\max_{\boldsymbol{a}} \sum_{n=1}^{N} b_{1,n}\mu_n\left(\boldsymbol{a}\right)R_n\left(t\right) - \max_{\boldsymbol{a}}\left[\sum_{n=1}^{N} b_{1,n}\mu_n\left(\boldsymbol{a}\right)\left(R_n\left(t\right) - R_n^e\right)\right] \underset{(a)}{\geq}$$

$$\sum_{n=1}^{N} b_{1,n}\mu_n\left(\boldsymbol{a}^*\left(t+1\right)\right)R_n\left(t\right) - \sum_{n=1}^{N} b_{1,n}\lambda_n E \geq \sum_{n=1}^{N} b_{1,n}\mu_n\left(\boldsymbol{a}^*\left(t+1\right)\right)R_n\left(t\right) - \lambda_{\max}E \quad (22)$$

where in (a) we used that $\mu_n\left(\boldsymbol{a}\right) \leq 1$ and that for all $n$, $R_n\left(t\right) - R_n^e \leq \lambda_n\left(t - T_e\right) \leq \lambda_n E$. Summing over all $t$ we obtain

$$\sum_{t=0}^{T-1}\sum_{n=1}^{N} b_{1,n}\mu_n\left(\boldsymbol{a}_{e(t)}^*\right)R_n^{e(t)} \geq \sum_{t=0}^{T-1}\sum_{n=1}^{N} b_{1,n}\mu_n\left(\boldsymbol{a}^*\left(t+1\right)\right)R_n\left(t\right) - \lambda_{\max}ET. \quad (23)$$

## 10  Proof of Lemma 3

At turn $t$ during epoch $e$, each agent knows the value of $\sum_{n=1}^{N} b_{1,n}R_n^e r_n\left(\boldsymbol{a}\left(t - 2D_E\right)\right)$, which serves as the effective reward of the team, thinking of the action profiles as the arms of a single-agent multi-armed bandit problem. Define $\mathcal{T}_e = \{T_e, \ldots, T_e + E - 1\}$. We define the expected effective reward of action profile $\boldsymbol{a}$ as

$$\mu\left(\boldsymbol{a}\right) = \sum_{n=1}^{N} b_{1,n}R_n^e\mu_n\left(\boldsymbol{a}\right). \quad (24)$$

Let $r_{\max} = \sum_{n=1}^{N} b_{1,n} R_n^e$, which bounds $\sum_{n=1}^{N} b_{1,n} R_n^e r_n (\boldsymbol{a}(t))$ for all $t$ in epoch $e$. With a slight abuse of notation, we use $v_{\boldsymbol{a}}^e(t)$ to denote the number of visits to action profile $\boldsymbol{a}$ during epoch $e$, and up to turn $t$. We also define the estimated expected reward of action profile $\boldsymbol{a}$ at turn $t > T_e + 2D_E$ as

$$\hat{\mu}_t (\boldsymbol{a}) = \frac{\sum_{\tau=T_e}^{t-2D_E-1} \mathbb{1}_{\{\boldsymbol{a}(\tau+1)=\boldsymbol{a}\}} \sum_{n=1}^{N} b_{1,n} R_n^e r_n (\boldsymbol{a}(\tau+1))}{v_{\boldsymbol{a}}^e (t-2D_E)}. \tag{25}$$

Let $\Delta_E = E^{-\frac{1}{2}} \log E$. Then the set of approximately optimal action profiles can also be written as

$$\mathcal{A}_e^* = \{\boldsymbol{a} \mid \mu(\boldsymbol{a}) \geq \mu(\boldsymbol{a}_e^*) - r_{\max}\Delta_E\}. \tag{26}$$

Define the regret of approximating $\boldsymbol{a}_e^*$ by

$$R_E \triangleq \sum_{t=T_e}^{T_e+E-1} \sum_{n=1}^{N} b_{1,n} R_n^e (\mu_n(\boldsymbol{a}_e^*) - r_n(\boldsymbol{a}(t+1))) \tag{27}$$

and note that this is **not** the QoS regret.

## 10.1 Taking a random subset of action profiles

Define the good event $A_G$ that happens when the subset $\mathcal{S}_E$ of $S_E = \lceil S_0 \log E \rceil$ action profiles that the agents pick includes an approximately optimal action profile $\boldsymbol{a} \in \mathcal{A}_e^*$. Note that $A_G$ is randomized at the beginning of epoch $e$ and depends on the QoS regret values $\{R_n^e\}$ that are given by $\mathcal{F}_e$. Then

$$\mathbb{P}(A_G^c \mid \mathcal{F}_e) \underset{(a)}{=} \left(1 - \frac{|\mathcal{A}_e^*|}{K^N}\right)^{\lceil S_0 \log E \rceil} = e^{\lceil S_0 \log E \rceil \log\left(1 - \frac{|\mathcal{A}_e^*|}{K^N}\right)} \leq E^{-\frac{S_0 |\mathcal{A}_e^*|}{K^N}} \tag{28}$$

where (a) follows since the action profiles in the subset are chosen independently and uniformly at random, so each has a probability of $\frac{|\mathcal{A}_e^*|}{K^N}$ to be approximately optimal. In the trivial case where $S_E = E \leq \lceil S_0 \log E \rceil$, the bound on $\mathbb{P}(A_G^c \mid \mathcal{F}_e)$ vanishes exponentially in $E$ instead. In the other trivial case where $S_E = K^N$, $\mathcal{S}_E$ is non-random and includes all action profiles, so $\mathbb{P}(A_G^c \mid \mathcal{F}_e) = 0$.

## 10.2 Regret of successive elimination with fixed delay

Algorithm 1 is a distributed algorithm in which all agents effectively run together a single-agent successive elimination on a random subset of action profiles as the set of arms, with a deterministic delay of $2D_E$ turns at epoch $e$. Next, we follow the proof of Theorem 2 in [29], with $q = 1$, with slight modifications. We note that the analysis in [29] holds for general stochastic delays, while there are only fixed delays in our case. We want to bound the regret of $\Delta_E$-optimal action profiles (i.e., arms) separately. The effective maximal reward of our successive elimination depends on the QoS regrets and is not bounded by a constant over epochs. Hence, using a bound of the form $\sum_{n=1}^{N} b_{1,n} R_n^e$ is crucial to our analysis since it can be bounded by $\sum_{n=1}^{N} b_{1,n} R_n(t)$, up to an approximation error, as we do in the proof of Theorem 2. However, since $\sum_{n=1}^{N} b_{1,n} R_n^e$ changes between epochs, the agents need to communicate to obtain it.

Define the confidence interval of action profile $\boldsymbol{a}$ as

$$\rho_{\boldsymbol{a}}(t) = \sqrt{\frac{2r_{\max}^2 \log E}{\max\{v_{\boldsymbol{a}}^e(t-2D_E), 1\}}} \tag{29}$$

and define the clean event as

$$A_c = \{|\hat{\mu}_t(\boldsymbol{a}) - \mu(\boldsymbol{a})| \leq \rho_{\boldsymbol{a}}(t), \forall \boldsymbol{a} \in \mathcal{S}_E, \forall t \in \mathcal{T}_e\}. \tag{30}$$

Let $\Delta_{\boldsymbol{a}} \triangleq \mu\left(\boldsymbol{a}_e^*\right) - \mu\left(\boldsymbol{a}\right)$, and let $R_{\boldsymbol{a}}$ be the accumulated regret from all the turns where agents played $\boldsymbol{a}$. We have

$$\mathbb{E}\left\{R_E \mid A_G, \mathcal{F}_e\right\} \le$$

$$\left(1 - \mathbb{P}\left(A_c\right)\right) r_{\max} E + \mathbb{P}\left(A_c\right) \sum_{\boldsymbol{a} \in \mathcal{S}_E} \mathbb{E}\left\{R_{\boldsymbol{a}} \mid A_c, A_G, \mathcal{F}_e\right\} \underset{(a)}{\le}$$

$$2 r_{\max} E^{-1} S_E + \sum_{\boldsymbol{a} \in \left(\mathcal{A}_e^*\right)^c \cap \mathcal{S}_E} \mathbb{E}\left\{R_{\boldsymbol{a}} \mid A_c, A_G, \mathcal{F}_e\right\} + \sum_{\boldsymbol{a} \in \mathcal{A}_e^* \cap \mathcal{S}_E} \mathbb{E}\left\{R_{\boldsymbol{a}} \mid A_c, A_G, \mathcal{F}_e\right\} \underset{(b)}{\le}$$

$$2 r_{\max} E^{-1} S_E + \sum_{\boldsymbol{a} \in \left(\mathcal{A}_e^*\right)^c \cap \mathcal{S}_E} \mathbb{E}\left\{\Delta_{\boldsymbol{a}} v_{\boldsymbol{a}}^e\left(T_E + E - 1\right) \mid A_c, A_G, \mathcal{F}_e\right\} + r_{\max} \Delta_E E \quad (31)$$

where (a) follows from Hoeffding's inequality on $1 - \mathbb{P}\left(A_c\right)$ (see [22], Page 158) along with a union bound on the $S_E$ action profiles. Inequality (b) follows by definition of $\mathcal{A}_e^*$.

Let $n_{\boldsymbol{a}}^e\left(t\right)$ be the number of times action profile $\boldsymbol{a}$ was played during epoch $e$ and between turn $t - 2D_E$ and turn $t$ (not including $t$). Then at turn $t$, if action profile $\boldsymbol{a}$ was visited $v_{\boldsymbol{a}}^e\left(t\right)$ times, we only know $v_{\boldsymbol{a}}^e\left(t\right) - n_{\boldsymbol{a}}^e\left(t\right)$ reward values due to the delay. Under the clean event $A_c$, an optimal action profile $\boldsymbol{a}_e^*$ is never eliminated. In the following, let $t \le T_E + E - 1$ be the last turn where a non-optimal action profile $\boldsymbol{a}$ was not eliminated. Then for this $t$

$$\mu\left(\boldsymbol{a}\right) + 2\rho_{\boldsymbol{a}}\left(t\right) \underset{(a)}{\ge} \hat{\mu}_t\left(\boldsymbol{a}\right) + \rho_{\boldsymbol{a}}\left(t\right) \underset{(b)}{\ge} \hat{\mu}_t\left(\boldsymbol{a}_e^*\right) - \rho_{\boldsymbol{a}_e^*}\left(t\right) \underset{(c)}{\ge} \mu\left(\boldsymbol{a}_e^*\right) - 2\rho_{\boldsymbol{a}_e^*}\left(t\right) \quad (32)$$

where (a) uses that under the clean event, $\mu\left(\boldsymbol{a}\right) \ge \hat{\mu}_t\left(\boldsymbol{a}\right) - \rho_{\boldsymbol{a}}\left(t\right)$ and (b) uses that action profile $\boldsymbol{a}$ was not eliminated. Inequality (c) uses that under the clean event $\mu\left(\boldsymbol{a}_e^*\right) \le \hat{\mu}_t\left(\boldsymbol{a}_e^*\right) + \rho_{\boldsymbol{a}_e^*}\left(t\right)$.

Hence

$$\Delta_{\boldsymbol{a}} = \mu\left(\boldsymbol{a}_e^*\right) - \mu\left(\boldsymbol{a}\right) \le 2\left(\rho_{\boldsymbol{a}}\left(t\right) + \rho_{\boldsymbol{a}_e^*}\left(t\right)\right) =$$

$$2\sqrt{\frac{2 r_{\max}^2 \log E}{\max\left\{v_{\boldsymbol{a}}^e\left(t - 2D_E\right), 1\right\}}} + 2\sqrt{\frac{2 r_{\max}^2 \log E}{\max\left\{v_{\boldsymbol{a}_e^*}^e\left(t - 2D_E\right), 1\right\}}} \underset{(a)}{\le}$$

$$4\sqrt{\frac{2 r_{\max}^2 \log E}{\max\left\{v_{\boldsymbol{a}}^e\left(t - 2D_E\right) - 1, 1\right\}}} \underset{(b)}{=} 4\sqrt{\frac{2 r_{\max}^2 \log E}{\max\left\{v_{\boldsymbol{a}}^e\left(t\right) - 1 - n_{\boldsymbol{a}}^e\left(t\right), 1\right\}}} \quad (33)$$

where (a) uses that $v_{\boldsymbol{a}_e^*}^e\left(t - 2D_E\right) \ge v_{\boldsymbol{a}}^e\left(t - 2D_E\right) - 1$ since we are going over all the non-eliminated action profiles in a round-robin manner, and (b) uses that $v_{\boldsymbol{a}}^e\left(t - 2D_E\right) = v_{\boldsymbol{a}}^e\left(t\right) - n_{\boldsymbol{a}}^e\left(t\right)$.

Therefore, (33) yields that for every non-optimal action profile $\boldsymbol{a}$

$$v_{\boldsymbol{a}}^e\left(T_E + E - 1\right) - n_{\boldsymbol{a}}^e\left(T_E + E - 1\right) - 1 \le \frac{32 r_{\max}^2 \log E}{\Delta_{\boldsymbol{a}}^2}. \quad (34)$$

Plugging in (34) in (31), we obtain

$$\mathbb{E}\left\{R_E \mid A_G, \mathcal{F}_e\right\} \le$$

$$2 r_{\max} E^{-1} S_E + \sum_{\boldsymbol{a} \in \left(\mathcal{A}_e^*\right)^c \cap \mathcal{S}_E} \left(\Delta_{\boldsymbol{a}} \frac{32 r_{\max}^2 \log E}{\Delta_{\boldsymbol{a}}^2} + \Delta_{\boldsymbol{a}}\left(n_{\boldsymbol{a}}^e\left(T_E + E - 1\right) + 1\right)\right) + r_{\max} \Delta_E E \underset{(a)}{\le}$$

$$r_{\max}\left(2 \frac{S_E}{E} + S_E \frac{32 \log E}{\Delta_E} + 2D_E + S_E + \Delta_E E\right) \underset{(b)}{\le}$$

$$\left(33\left(S_0 + 1\right) \log E \sqrt{E} + 2D_E\right) \sum_{n=1}^N b_{1,n} R_n^e \quad (35)$$

where (a) uses $\Delta_{\boldsymbol{a}} \ge r_{\max} \Delta_E$ for all $\boldsymbol{a} \in \left(\mathcal{A}_e^*\right)^c$, and also that $\Delta_{\boldsymbol{a}} \le r_{\max}$ and then that $\sum_{\boldsymbol{a} \in \left(\mathcal{A}_e^*\right)^c \cap \mathcal{S}_E} n_{\boldsymbol{a}}^e\left(T_E + E - 1\right) \le 2D_E$ which follows since we played exactly $2D_E$ action profiles between turn $T_E + E - 1 - 2D_E$ and turn $T_E + E - 1$, and in the worst case they are all in $\left(\mathcal{A}_e^*\right)^c$. Inequality (b) uses that $r_{\max} = \sum_{n=1}^N b_{1,n} R_n^e$, $2 \frac{S_E}{E} + S_E \le \sqrt{E} S_E$ for $E \ge 3$ and $S_E \le S_0 \log E + 1$.

## 10.3 Concluding the proof

We can now conclude by bounding the expected regret $R_E$ as follows

$$\mathbb{E}\left\{\sum_{t=T_e}^{T_e+E-1}\sum_{n=1}^{N} b_{1,n}R_n^e\left(\mu_n\left(\boldsymbol{a}_e^*\right)-r_n\left(\boldsymbol{a}\left(t+1\right)\right)\right) \mid \mathcal{F}_e\right\}=$$

$$\mathbb{E}\left\{R_E \mid A_G,\mathcal{F}_e\right\}\mathbb{P}\left(A_G \mid \mathcal{F}_e\right)+\mathbb{E}\left\{R_E \mid A_G^c,\mathcal{F}_e\right\}\mathbb{P}\left(A_G^c \mid \mathcal{F}_e\right)\underset{(a)}{\leq}$$

$$\left(33\left(S_0+1\right)\log E\sqrt{E}+2D_E+E^{1-\frac{S_0|\mathcal{A}_e^*|}{KN}}\right)\sum_{n=1}^{N} b_{1,n}R_n^e \quad (36)$$

where in (a) we used that $\mathbb{E}\left\{R_E \mid A_G^c,\mathcal{F}_e\right\}\leq\left[\sum_{n=1}^{N} b_{1,n}R_n^e\right]E$, the bound in (28), and (35).

# 11 Proof of Theorem 2

Throughout the proof, we use $E_T$ to denote the epoch length, to emphasize that it can depend on $T$. We want to bound the total drift of the Lyapunov function, $\mathbb{E}\left\{L\left(T\right)-L\left(0\right)\right\}$. Define the filtration

$$\mathcal{F}\left(t\right)=\sigma\left(\left\{\boldsymbol{a}\left(\tau\right),\left\{z_n\left(\tau\right)\right\}_{n\in\mathcal{N}} \mid \tau\leq t\right\}\right). \quad (37)$$

Hence $R_n\left(t\right)$ is $\mathcal{F}\left(t\right)$-measurable for all $n$ and $t$. Let

$$\varepsilon_{E_T}=\frac{33\left(S_0+1\right)\log E_T}{\sqrt{E_T}}+2\frac{D_{E_T}}{E_T}+E_T^{-\frac{S_0\min_{\boldsymbol{\alpha}\in\Delta^N}|\mathcal{A}^*(\boldsymbol{\alpha})|}{KN}} \quad (38)$$

where $\min_{\boldsymbol{\alpha}\in\Delta}|\mathcal{A}^*\left(\boldsymbol{\alpha}\right)|$ is the minimal number of action profiles that approximately maximize $\sum_{n=1}^{N}\alpha_n\mu_n\left(\boldsymbol{a}\right)$, taken over all $\boldsymbol{\alpha}$ in the $N$-dimensional simplex (see (4)).

Recall that

$$\delta\left(\boldsymbol{\lambda},E_T\right)=\max_{\boldsymbol{p}\in\Delta^{K^N}}\min_n\left(\sum_{\boldsymbol{a}} p\left(\boldsymbol{a}\right)\mu_n\left(\boldsymbol{a}\right)-\lambda_n-\varepsilon_{E_T}\right). \quad (39)$$

## 11.1 Proof Sketch

Lemma 3 shows that Algorithm 1 incurs an error of $\varepsilon_E E\sum_{n=1}^{N} b_{1,n}R_n^e$ for some $\varepsilon_E$ that vanishes with $E$. However, an error of $\varepsilon_E E\sum_{n=1}^{N} b_{1,n}R_n^e$ depends on the QoS regrets $R_n\left(t\right)$. Hence, we bound $\varepsilon_E E\sum_{n=1}^{N} b_{1,n}R_n^e$ with $\varepsilon_E\sum_{n=1}^{N} b_{1,n}\left(1-\lambda_n\right)E^2+\sum_{t=T_e}^{T_e+E-1}\sum_{n=1}^{N} b_{1,n}\varepsilon_{E_T}R_n\left(t\right)$, which follows since in the worst case $R_n^e$ is lagging behind $R_n\left(t\right)$ by $\left(1-\lambda_n\right)E$ for all $t$ in epoch $e$. Summing all the errors, we get the bound in (7) on $\sum_{t=0}^{T-1}\sum_{n=1}^{N} b_{1,n}\mathbb{E}\left\{R_n\left(t\right)\right\}$. For $\boldsymbol{\lambda}\in C_{\varepsilon_E}\left(\mathcal{G}\right)$, this bound implies that all agents have an $O\left(1\right)$ empirical average expected QoS regret, over $t=1,\ldots,T$. We then prove that if $\boldsymbol{\lambda}\in C_{\varepsilon_E}\left(\mathcal{G}\right)$, then $\mathbb{E}\left\{R_n\left(t\right)\right\}=O\left(\sqrt{E_T t}\right)$ for all $t$ and $n$, by arguing that if $\mathbb{E}\left\{R_n\left(t\right)\right\}$ is too large for any $t$ and $n$, it would violate the bound on $\sum_{t=0}^{T-1}\sum_{n=1}^{N} b_{1,n}\mathbb{E}\left\{R_n\left(t\right)\right\}$ since $\mathbb{E}\left\{R_n\left(t-\tau\right)\right\}$ for the recent turns is also large. Finally, by taking $E_T$ to be increasing sublinearly with $T$, the latter bound implies that for large enough $T$, we have for all $\boldsymbol{\lambda}\in C\left(\mathcal{G}\right)$ both $\frac{1}{T}\sum_{t=0}^{T-1}\mathbb{E}\left\{R_n\left(t\right)\right\}=O\left(E_T\right)$ for all $n$ and $\mathbb{E}\left\{R_n\left(t\right)\right\}=O\left(\sqrt{t E_T}\right)$ for all $t$ and $n$.

## 11.2 Lower bound on the weighted sum of rewards of a max-weight matching

Let $\lambda_{\min},\lambda_{\max}$ be the minimal and maximal values of $\boldsymbol{\lambda}$, respectively. If $\left(\lambda_1,\ldots,\lambda_N\right)\in C_{\varepsilon_{E_T}}\left(\mathcal{G}\right)$ then $\delta\left(\boldsymbol{\lambda},E_T\right)>0$, and by definition there exists a policy that assigns a probability function $p\left(\boldsymbol{a}\right)$ to all action profiles $\boldsymbol{a}\in\mathcal{A}$ such that for all $n$:

$$\mathbb{E}^{\boldsymbol{P}}\left\{\mu_n\left(\boldsymbol{a}\right)\right\}\geq\lambda_n+\delta\left(\boldsymbol{\lambda},E_T\right)+\varepsilon_{E_T}. \quad (40)$$

However, the max-weight matching $\boldsymbol{a}^*(t+1) = \arg\max_{\boldsymbol{a}} \sum_{n=1}^{N} b_{1,n} R_n(t) \mu_n(\boldsymbol{a})$ minimizes the drift among all policies, even if they take into account $R_n(t)$. Hence

$$\sum_{n=1}^{N} b_{1,n} R_n(t) \mathbb{E}\{\mu_n(\boldsymbol{a}^*(t+1)) \mid \mathcal{F}(t)\} \geq \sum_{n=1}^{N} b_{1,n} R_n(t) \mathbb{E}^{\boldsymbol{p}}\{\mu_n(\boldsymbol{a}) \mid \mathcal{F}(t)\} \underset{(a)}{=}$$

$$\sum_{n=1}^{N} b_{1,n} R_n(t) \mathbb{E}^{\boldsymbol{p}}\{\mu_n(\boldsymbol{a})\} \geq \sum_{n=1}^{N} (\lambda_n + \delta(\boldsymbol{\lambda}, E_T) + \varepsilon_{E_T}) b_{1,n} R_n(t) \quad (41)$$

where (a) follows since $\boldsymbol{p}$ is independent of the past actions (and the QoS regrets).

## 11.3    Additive error of replacing $R_n(t)$ by $R_n^{e(t)}$

The QoS regret of agent $n$ at turn $t$ cannot be too far from its QoS regret at the beginning of epoch $e = e(t)$:

$$R_n(t) \geq R_n^e - \sum_{\tau=T_e}^{t} (r_n(\boldsymbol{a}(\tau)) - \lambda_n) \geq R_n^e - (1 - \lambda_n) E_T, \quad (42)$$

so by summing over all turns in epoch $e$, we obtain

$$\sum_{t=T_e}^{T_e+E_T-1} b_{1,n} R_n(t) \geq b_{1,n} E_T R_n^e - b_{1,n} (1 - \lambda_n) E_T^2 \quad (43)$$

which yields

$$b_{1,n} R_n^e \varepsilon_{E_T} E_T \leq b_{1,n} (1 - \lambda_n) \varepsilon_{E_T} E_T^2 + \sum_{t=T_e}^{T_e+E_T-1} b_{1,n} \varepsilon_{E_T} R_n(t). \quad (44)$$

Since $0 \leq r_n(\boldsymbol{a}(t)) \leq 1$, we get from (42) that

$$\sum_{t=0}^{T-1} \sum_{n=1}^{N} b_{1,n} \mathbb{E}\left\{ r_n(\boldsymbol{a}(t+1)) \left( R_n(t) - R_n^{e(t)} \right) \right\} \geq$$

$$-E_T \sum_{t=0}^{T-1} \sum_{n=1}^{N} b_{1,n} (1 - \lambda_n) \mathbb{E}\{r_n(\boldsymbol{a}(t+1))\} \geq -(1 - \lambda_{\min}) E_T T. \quad (45)$$

## 11.4    Bounding the sum of QoS regrets

We can now bound from below the expected weighted reward as follows:

$$\sum_{t=0}^{T-1} \sum_{n=1}^{N} b_{1,n} \mathbb{E}\{R_n(t) r_n(\boldsymbol{a}(t+1))\} =$$

$$\sum_{t=0}^{T-1} \sum_{n=1}^{N} \left( b_{1,n} \mathbb{E}\left\{ R_n^{e(t)} r_n(\boldsymbol{a}(t+1)) \right\} + b_{1,n} \mathbb{E}\left\{ r_n(\boldsymbol{a}(t+1)) \left( R_n(t) - R_n^{e(t)} \right) \right\} \right) \underset{(a)}{\geq}$$

$$\sum_{t=0}^{T-1} \sum_{n=1}^{N} b_{1,n} \mathbb{E}\left\{ \mu_n\left(\boldsymbol{a}_{e(t)}^*\right) R_n^{e(t)} \right\} - (1 - \lambda_{\min}) E_T T - \varepsilon_{E_T} E_T \sum_{e=1}^{\left\lceil \frac{T}{E_T} \right\rceil} \sum_{n=1}^{N} b_{1,n} \mathbb{E}\{R_n^e\} \underset{(b)}{\geq}$$

$$\sum_{t=0}^{T-1} \sum_{n=1}^{N} b_{1,n} \mathbb{E}\{R_n(t) \mu_n(\boldsymbol{a}^*(t+1))\} - (1 + \lambda_{\max} - \lambda_{\min}) E_T T - \varepsilon_{E_T} E_T \sum_{e=1}^{\left\lceil \frac{T}{E_T} \right\rceil} \sum_{n=1}^{N} b_{1,n} \mathbb{E}\{R_n^e\}$$

$$(46)$$

where (a) uses (45) and Lemma 3 with the tower rule on $\mathbb{E}\left\{\cdot \mid \mathcal{F}_{e(t)}\right\}$. We also used the definition of $\varepsilon_{E_T}$ in (38), along with the fact that $\left|\mathcal{A}_e^*\right| \geq \min_{\boldsymbol{\alpha} \in \Delta}\left|\mathcal{A}^*\left(\boldsymbol{\alpha}\right)\right|$ for all $e$. Inequality (b) uses Lemma 2.

Next, by taking the expectation on Lemma 1 we obtain

$$
\mathbb{E}\left\{L\left(T\right) - L\left(0\right)\right\} = \sum_{t=0}^{T-1} \mathbb{E}\left\{L\left(t+1\right) - L\left(t\right)\right\} \leq
$$

$$
2T + 2\sum_{t=0}^{T-1}\sum_{n=1}^{N} \lambda_n b_{1,n} \mathbb{E}\left\{R_n\left(t\right)\right\} - 2\sum_{t=0}^{T-1}\sum_{n=1}^{N} b_{1,n}\mathbb{E}\left\{R_n\left(t\right) r_n\left(\boldsymbol{a}\left(t+1\right)\right)\right\} \underset{(a)}{\leq}
$$

$$
2T + 2\sum_{t=0}^{T-1}\sum_{n=1}^{N} \lambda_n b_{1,n}\mathbb{E}\left\{R_n\left(t\right)\right\} - 2\sum_{t=0}^{T-1}\sum_{n=1}^{N} b_{1,n}\mathbb{E}\left\{R_n\left(t\right) \mu_n\left(\boldsymbol{a}^*\left(t+1\right)\right)\right\}
$$

$$
+ 2\left(1 + \lambda_{\max} - \lambda_{\min}\right) E_T T + 2 E_T \varepsilon_{E_T} \sum_{e=1}^{\left\lceil \frac{T}{E_T} \right\rceil}\sum_{n=1}^{N} b_{1,n}\mathbb{E}\left\{R_n^e\right\} \underset{(b)}{\leq}
$$

$$
2T - 2\left(\delta\left(\boldsymbol{\lambda}, E_T\right) + \varepsilon_{E_T}\right)\sum_{t=0}^{T-1}\sum_{n=1}^{N} b_{1,n}\mathbb{E}\left\{R_n\left(t\right)\right\}
$$

$$
+ 2\left(1 + \lambda_{\max} - \lambda_{\min}\right) E_T T + 2 E_T \varepsilon_{E_T} \sum_{e=1}^{\left\lceil \frac{T}{E_T} \right\rceil}\sum_{n=1}^{N} b_{1,n}\mathbb{E}\left\{R_n^e\right\} \underset{(c)}{\leq}
$$

$$
2T - 2\delta\left(\boldsymbol{\lambda}, E_T\right)\sum_{t=0}^{T-1}\sum_{n=1}^{N} b_{1,n}\mathbb{E}\left\{R_n\left(t\right)\right\} + 2\left(1 + \lambda_{\max} - \lambda_{\min}\right) E_T T + 2\left(1 - \lambda_{\min}\right) E_T T \varepsilon_{E_T}
$$

$$
\tag{47}
$$

where (a) uses (46), (b) uses (41) and (c) uses (44).

Next, (47) implies

$$
2\delta\left(\boldsymbol{\lambda}, E_T\right)\sum_{t=0}^{T-1}\sum_{n=1}^{N} b_{1,n}\mathbb{E}\left\{R_n\left(t\right)\right\} \leq
$$

$$
2T + 2\left(1 + \lambda_{\max} - \lambda_{\min}\right) E_T T + 2\left(1 - \lambda_{\min}\right)\varepsilon_{E_T} E_T T + \mathbb{E}\left\{L\left(0\right) - L\left(T\right)\right\} \underset{(a)}{\leq}
$$

$$
2T + 2\left(1 + \lambda_{\max} - \lambda_{\min}\right) E_T T + 2\left(1 - \lambda_{\min}\right)\varepsilon_{E_T} E_T T \tag{48}
$$

where (a) uses that $L\left(0\right) = 0$ and $L\left(T\right) \geq 0$.

We conclude that if $\left(\lambda_1, \ldots, \lambda_N\right) \in C_{\varepsilon_{E_T}}\left(\mathcal{G}\right)$ then

$$
\frac{1}{T}\sum_{t=0}^{T-1}\sum_{n=1}^{N} b_{1,n}\mathbb{E}\left\{R_n\left(t\right)\right\} \leq \frac{1 + \left(\left(1 + \lambda_{\max} - \lambda_{\min}\right) + \varepsilon_{E_T}\left(1 - \lambda_{\min}\right)\right) E_T}{\delta\left(\boldsymbol{\lambda}, E_T\right)}. \tag{49}
$$

## 11.5 Sublinear Expected QoS Regret

For $t \leq E_T$ we have $\mathbb{E}\left\{R_n\left(t\right)\right\} \leq \lambda_n t \leq \lambda_n \sqrt{E_T t}$ for all $n$. Now, assume that for some $t > E_T$ and some $n$ we have that $\mathbb{E}\left\{R_n\left(t\right)\right\} \geq M$ for some $M > 0$. Then, using that $R_n\left(t\right) - R_n\left(t-1\right) \leq \lambda_n$, we know that

$$
\mathbb{E}\left\{R_n\left(t-1\right)\right\} = \mathbb{E}\left\{R_n\left(t\right)\right\} + \mathbb{E}\left\{R_n\left(t-1\right) - R_n\left(t\right)\right\} \geq M - \lambda_n \tag{50}
$$

and iterating over (50) for $t - \tau$ times we get that for $\tau \geq t - \left\lfloor \frac{M}{\lambda_n} \right\rfloor$

$$
\mathbb{E}\left\{R_n\left(\tau\right)\right\} \geq M - \left(t - \tau\right)\lambda_n. \tag{51}
$$

Therefore,

$$\sum_{\tau=0}^{t} \mathbb{E}\{R_n(\tau)\} \geq \sum_{\tau=t-\lfloor\frac{M}{\lambda_n}\rfloor}^{t} \mathbb{E}\{R_n(\tau)\} \geq M\left(\left\lfloor\frac{M}{\lambda_n}\right\rfloor + 1\right) - \lambda_n \sum_{\tau=0}^{\lfloor\frac{M}{\lambda_n}\rfloor} \tau =$$

$$M\left(\left\lfloor\frac{M}{\lambda_n}\right\rfloor + 1\right) - \lambda_n \frac{\left\lfloor\frac{M}{\lambda_n}\right\rfloor\left(\left\lfloor\frac{M}{\lambda_n}\right\rfloor + 1\right)}{2} = \left(\left\lfloor\frac{M}{\lambda_n}\right\rfloor + 1\right)\left(M - \frac{\lambda_n}{2}\left\lfloor\frac{M}{\lambda_n}\right\rfloor\right) \geq \frac{M^2}{2\lambda_n}. \quad (52)$$

We know that if $(\lambda_1, \ldots, \lambda_N) \in C_{\varepsilon_{E_T}}(\mathcal{G})$, then $\delta(\boldsymbol{\lambda}, E_T) > 0$ and, by (48), there exists a constant $B_0 > 0$ such that

$$b_{1,n} \sum_{\tau=0}^{t} \mathbb{E}\{R_n(\tau)\} \leq \sum_{\tau=0}^{t}\sum_{m=1}^{N} b_{1,m}\mathbb{E}\{R_m(\tau)\} \leq B_0 E_T t \quad (53)$$

where we used that $0 < b_{1,m} \leq 1$ for all $m$, which only depends on the communication matrix $W$ and $D_E$. We also used that $\varepsilon_{E_T}$ decreases with $T$, and hence $\delta(\boldsymbol{\lambda}, E_T)$ increases with $T$. Thus, $\frac{E_T t}{\delta(\boldsymbol{\lambda}, E_T)} = O(E_T t)$.

Hence, (52) implies that for any $M$ such that $\frac{M^2}{2\lambda_n} > \frac{B_0}{b_{1,n}} E_T t$, the assumption that $\mathbb{E}\{R_n(t)\} \geq M$ contradicts (53). Thus, we must have

$$\frac{M^2}{2\lambda_n} \leq \frac{B_0}{b_{1,n}} E_T t \implies M \leq \sqrt{\frac{2\lambda_n B_0}{b_{1,n}} E_T t}. \quad (54)$$

Therefore, $\mathbb{E}\{R_n(t)\} \leq B_1\sqrt{E_T t}$ for some constant $B_1 > 0$, and so for all $T$ and every $n$,

$$\frac{\mathbb{E}\{R_n(T)\}}{T} \leq B_1\sqrt{\frac{E_T}{T}} \quad (55)$$

which, if $\frac{E_T}{T} \to 0$ as $T \to \infty$, implies that

$$\lim_{T\to\infty}\frac{\mathbb{E}\{R_n(T)\}}{T} = 0. \quad (56)$$

If $E_T = E$ instead (i.e., the epoch size is constant with respect to $T$), and $(\lambda_1, \ldots, \lambda_N) \in C_{\varepsilon_E}(\mathcal{G})$, then by taking the limit of (53) for $t = T - 1$ and summing over $n$, we conclude that

$$\limsup_{T\to\infty}\frac{1}{T}\sum_{t=0}^{T-1}\sum_{n=1}^{N} \mathbb{E}\{R_n(t)\} < \sum_{n=1}^{N}\frac{B_0 E}{b_{1,n}} < \infty. \quad (57)$$

## 12 Time-Varying Stochastic Communication Graph

In this section, we extend our results to the case of a time-varying communication graph. In practice, agent mobility, obstacles, and link failures all induce a time-varying communication graph that hinders the coordination between the agents, so proving QoS regret guarantees for this case is important to establish the robustness of our distributed algorithm.

This scenario requires slight modifications to our algorithm, as detailed in Algorithm 2. Obviously, Algorithm 2 can be used in the special case of a fixed communication graph as well. However, for a fixed communication graph, the tuning of Algorithm 1 results in better performance overall.

Instead of a fixed communication graph, we assume the following:

**Definition 5.** We consider a sequence of graphs $G(t) = (\mathcal{N}, E(t))$ where the set of edges $E(t)$ is stochastic and i.i.d. over time, and $G(t)$ takes values in $\{G_1, \ldots, G_M\}$ for some $M > 0$. We assume that $\bigcup_{i=1}^{M} G_i$ is connected, where the union is over the sets of edges. The communication matrix at turn $t$ is $W(t) = W_i$ if $G(t) = G_i$, such that $w_{i,n,m} > 0$ if and only if $G_{i,n,m} = 1$ and $\sum_{m=1}^{N} w_{i,n,m} = 1$ for all $n$. We also assume that the sequence $G(t)$ is independent of the sequence $z_n(t) = r_n(\boldsymbol{a}(t)) - \mu_n(\boldsymbol{a}(t))$ for all $n$.

Extending our result to this case of time-varying stochastic graphs involves several interesting technical issues. First, with a stochastic communication graph, the coefficients in our weighted sum of rewards are no longer constant. We define $b_{1,n}(t)$ as the $n$-th element in the first row of $\prod_{\tau=t-2D_E+1}^{t-D_E} W(\tau)$. Since $W(t)$ is i.i.d., we also use the notation $\mathbb{E}\{b_{1,n}\} = \mathbb{E}\{b_{1,n}(t)\}$ to emphasize that $\mathbb{E}\{b_{1,n}(t)\}$ is fixed with time. While $G(t)$ is i.i.d. and therefore memoryless, our weighted sum of rewards $\sum_{n=1}^{N} b_{1,n}(t+2D_E) R_n(t) r_n(\boldsymbol{a}(t+1))$ is no longer memoryless. This follows since $b_{1,n}(t)$ and $b_{1,n}(t-1)$ both depend on $G(t-D_E-1), \ldots, G(t-2D_E+1)$. As a result, bounding the error probability of the clean event, where all the estimated rewards are within their confidence intervals, requires special attention. We use the fact that $b_{1,n}(t)$ is independent of $b_{1,n}(t-D_E)$ to obtain an alternative error bound, that requires a larger confidence interval.

To obtain a bound on the maximal effective reward $\sum_{n=1}^{N} b_{1,n}(t+2D_E) R_n(t) r_n(\boldsymbol{a}(t+1))$ of the successive elimination step, agents in Algorithm 2 propagate the maximal initial QoS regret $R_m^e$ they received from others, for the first $D_E$ turns of every epoch. Agents append the symbol '-1' to $R_m^e$ to distinguish these messages from the consensus averaging messages.

With our time-varying communication graph model, $G(t)$ does not have to be connected and may even be disconnected for every $t$. This fact complicates the communication step of Algorithm 2, which can now fail for three different reasons, as we detail next.

*Remark* 1. For the communication step of Algorithm 2 to be successful, the following conditions have to hold:

1. $\mathbb{E}\{b_{1,n}(t)\} \geq \kappa_0 > 0$ for all $n$, for a constant $\kappa_0$ that is independent of $E$. Without this, our expected QoS regret bound does not include the QoS regrets of all agents.

2. The messages from agent 1 will reach all other agents in no more than $D_E$ turns with high probability. Without this, agents do not agree on their effective rewards and do not compute the same estimated expectations, LCB or UCB for the "arms".

3. $\tilde{R}_n^e = \tilde{R}_m^e$ for all $n, m$. At the beginning of epoch $e$, agents agree on the value of $r_{\max} = \max_n R_n^e$ that they all need to compute their confidence radius.

To bound the regret in learning an approximately optimal weighted sum of rewards, we have to prove that $\mathbb{E}\{b_{1,n}(t)\} > 0$ for all $n$ and to bound the probability of the two failure events above. The analysis suggests that we now have to pick $D_E = \lceil S_1 \log^2 E \rceil$, as opposed to the constant $D_E = d(G)$ that is sufficient for a fixed communication graph. This makes the communication overhead of Algorithm 2 $O(\log^2 E)$ per agent per turn. This increased communication overhead reflects the fact that with a time-varying graph that does not have to be connected every turn, in the worst case more attempts and turns are needed to deliver the same amount of information.

Finally, we need to redefine the Lyapunov function, which now has time-varying coefficients.

**Definition 6.** Define the matrix $B(t) = \prod_{\tau=t-2D_E+1}^{t-D_E} W(\tau)$, and denote its $(i,j)$ element by $b_{i,j}(t)$. Define the following Lyapunov function such that $L(0) = 0$ and for all $t \geq 1$

$$L(t) \triangleq \sum_{n=1}^{N} b_{1,n}(t + 2D_E) R_n^2(t). \tag{58}$$

We prove the following result, generalizing Theorem 2.

**Theorem 3.** *Let $\mathcal{G}$ be a multiplayer multi-armed bandit game, played for $T$ turns. Let $\boldsymbol{\lambda} \in C(\mathcal{G})$. Let the agents run Algorithm 2 with epochs of length $E$, a random subset of action profiles of size $S_E \leq \lceil S_0 \log E \rceil$ for some $S_0 > 0$ and a delay of $D_E = \lceil S_1 \log^2 E \rceil$ for some $S_1 > 0$ such that $D_E \geq N$. Assume that $G(t)$ is an i.i.d. sequence taking values in $\{G_1, \ldots, G_M\}$ such that $\bigcup_{i=1}^{M} G_i$ is connected. Assume that there is a communication matrix $W_i$ corresponding to $G_i$ for every $i$ (see Definition 5). Define the set of approximately optimal action profiles*

$$\mathcal{A}^*(\boldsymbol{\alpha}) = \left\{ \boldsymbol{a} \mid \sum_{n=1}^{N} \alpha_n \mu_n(\boldsymbol{a}) \geq \max_{\boldsymbol{a}'} \sum_{n=1}^{N} \alpha_n \mu_n(\boldsymbol{a}') - \log E \sqrt{\frac{D_E}{E}} \right\} \tag{59}$$

*and*

$$\varepsilon_E = \frac{1}{\kappa_1} \left[ \frac{\sqrt{2S_1}(64S_0 + 1)\log^2 E}{\sqrt{E}} + \frac{14S_0 S_1 \log^3 E}{E} \right.$$
$$\left. + E^{1-S_1\kappa_0 \log E} + E^{-\frac{S_0 \min_{\boldsymbol{\alpha} \in \Delta^N} |\mathcal{A}^*(\boldsymbol{\alpha})|}{K^N}} \right] \tag{60}$$

*for some constants $\kappa_0, \kappa_1 > 0$ that depend on the distribution of $W(t)$. Define*

$$\delta(\boldsymbol{\lambda}, E) = \max_{\boldsymbol{p} \in \Delta^{K^N}} \min_n \left( \sum_{\boldsymbol{a}} p(\boldsymbol{a}) \mu_n(\boldsymbol{a}) - \lambda_n - \varepsilon_E \right). \tag{61}$$

*If $\boldsymbol{\lambda} \in C_{\varepsilon_E}(\mathcal{G})$ then the QoS regrets (Definition 2) satisfy*

$$\frac{1}{T} \sum_{t=0}^{T-1} \sum_{n=1}^{N} \mathbb{E}\{b_{1,n}\} \mathbb{E}\{R_n(t)\} \leq \frac{1 + (2 + \varepsilon_{E_T})\lambda_{\max} E_T}{\delta(\boldsymbol{\lambda}, E_T)} \tag{62}$$

*where $\mathbb{E}\{\cdot\}$ is over the rewards, the communication graphs, and the random subsets, and $\mathbb{E}\{b_{1,n}\} \geq \kappa_0 > 0$ for all $n$, for a constant $\kappa_0$ that is independent of $E$.*

*Therefore, for any $\boldsymbol{\lambda} \in C_{\varepsilon_E}(\mathcal{G})$, Algorithm 2 achieves $O(1)$ empirical average expected QoS regret for all $n$, i.e., $\limsup_{T \to \infty} \frac{1}{T} \sum_{t=0}^{T-1} \sum_{n=1}^{N} \mathbb{E}\{R_n(t)\} < \infty$ (strong stability) and $\mathbb{E}\{R_n(t)\} = O(\sqrt{t})$ for all $n, t$. Furthermore, if we pick $E_T$ such that $E_T \to \infty$ and $\frac{E_T}{T} \to 0$ as $T \to \infty$, then for any $\boldsymbol{\lambda} \in C(\mathcal{G})$, Algorithm 2 achieves $\frac{1}{T} \sum_{t=0}^{T-1} \sum_{n=1}^{N} \mathbb{E}\{R_n(t)\} = O(E_T)$ for all $n$ and $\mathbb{E}\{R_n(t)\} = O(\sqrt{E_T t})$ for all $n, t$ (mean rate stability).*

**Algorithm 2** Dynamic Multiplayer Bandit QoS Learning (time-varying communication graph)

**Initialization:** Let $e, l = 0$. Let $W(t)$ be a sequence of communication matrices that follows Definition 5. Let $S_E = \min\left\{\lceil S_0 \log E \rceil, K^N, E\right\}$ and $D_E = \lceil S_1 \log^2 E \rceil$ for some $S_0, S_1 > 0$.
**Input:** $\lambda_n \geq 0$ and agent index $n$
**For each** $e \geq 0$**, each agent** $n$ **runs:**

1. Compute $R_n^e = \max\left\{0, \lambda_n t - \sum_{\tau=1}^{t} r_n(\boldsymbol{a}(\tau))\right\}$.          ▷ QoS regret for epoch $e$

2. Randomize actions $a_n^1, \ldots, a_n^{S_E}$, uniformly and independently over $\{0, \ldots, K-1\}$. Store their indices in the ordered set $\mathcal{B} = \{1, \ldots, S_E\}$, where $\mathcal{B}(l)$ is the $l$-th element.

3. Set $v_i = \hat{\mu}_i = 0$ for all $i = 1, \ldots, S_E$ and $\tilde{R}_{\max,n} = R_n^e$.

4. **For** $E$ turns (from $t = T_e$ to $t = T_e + E - 1$):

    (a) Play $a_n(t) = a_n^{\mathcal{B}(l)}$ and receive reward $r_n(\boldsymbol{a}(t))$.        ▷ action step

    (b) If $T_e \leq t \leq T_e + D_E - 1$ then receive $\left(\tilde{R}_{\max,m}, -1\right)$ from all neighbors $m \in \mathcal{N}_n$, set $\tilde{R}_{\max,n} = \max_{m \in \mathcal{N}_n} \tilde{R}_{\max,m}$ and broadcast $\left(\tilde{R}_{\max,n}, -1\right)$.     ▷ $r_{max}$ initialization

    (c) Set $\theta_n^0 = r_n(\boldsymbol{a}(t)) R_n^e$ and $\theta_m^0 = 0$ for all $m \neq n$.

    (d) Broadcast $\{(\theta_n^\tau, \tau)\}_{\tau=0}^{D_E-1}$ to neighbors and receive $\{(\theta_m^\tau, \tau)\}_{\tau=0}^{D_E-1}$ from neighbors.

    (e) Set $\theta_n^{\tau+1} = \sum_{m=1}^{N} w_{n,m}(t) \theta_m^\tau, \forall \tau = 0, \ldots, D_E - 1$. ▷ $D_E$ parallel consensus steps

    (f) If you know $\theta_1^\tau$ for $D_E \leq \tau \leq 2D_E - 1$ then broadcast $(\theta_1^\tau, \tau)$. ▷ reaching agreement

    (g) If you receive $(\theta_1^\tau, \tau)$ for $D_E \leq \tau \leq 2D_E - 1$ then set $S(t - \tau) = \theta_1^\tau$.

    (h) Let $k = \mathcal{B}(l - 2D_E)$ and update $v_k \leftarrow v_k + 1$.

    (i) Compute the confidence radius $\rho_k = \sqrt{\frac{2\tilde{R}_{\max,n}^2 D_E \log E}{\max\{v_k, 1\}}}$ and

    $$\hat{\mu}_k \leftarrow \left(1 - \frac{1}{v_k}\right)\hat{\mu}_k + \frac{S(t - 2D_E)}{v_k} \quad, \quad L_k = \hat{\mu}_k - \rho_k \quad, \quad U_k = \hat{\mu}_k + \rho_k$$

    (j) If $l = |\mathcal{B}|$ then delete from $\mathcal{B}$ all $i$ such that $U_i < \max_{j \in \mathcal{B}} L_j$, and set $l = 0$. ▷ elimination

    (k) update $l \leftarrow l + 1$.

5. Update $e \leftarrow e + 1$

**End**

## 12.1 Proof of Theorem 3

In the following, we explain how to modify the proofs of Lemma 3 and Theorem 2 for stochastic time-varying communication graphs. Note that Lemma 1 still holds with $b_{1,n}(t + 2D_E)$ replacing $b_{1,n}$, and Lemma 2 holds with $\mathbb{E}\{b_{1,n}(t)\}$ replacing $b_{1,n}$.

### 12.1.1 Communication failure events

Define $\mathcal{T}_e = [T_e, T_e + E - 1]$. The two communication failure events detailed in Remark 1 are avoided if a message from agent 1 reaches all other agents in no more than $D_E$ turns, and if agent 1 receives a message from all other agents in no more than $D_E$ turns. For the learning in the epoch to succeed, we have to avoid a communication failure of one of the two types for almost $E$ times - once for the initialization of $r_{\max} = \max_n R_n^e$, and $E - 2D_E$ times for the first $E - 2D_E$ turns for which the agents are able to obtain $\sum_{n=1}^{N} b_{1,n}(t + 2D_E) R_n(t) r_n(\boldsymbol{a}(t+1))$ before the end of the epoch. We define $S_C$ as the communication success event for epoch $e$, where none of these communication failures occurred.

Let $W_i$ be the communication matrix associated with $G_i$ (see Definition 5). Since $\bigcup_{i=1}^{M} G_i$ is connected and each $G_i$ has self-loops, it means that $\bigcup_{i=1}^{M} G_i$ has a path of length $N$ between any two of

the $N$ nodes. Hence, there exists a sequence of length $N^3$ such that $\prod_{j=1}^{N^3} W_{p(j)}$ has only positive elements, where $p(j) \in \{1, \ldots, M\}$ is the type of the $j$-th graph in the sequence. Starting from turn $t$, there is some probability $p_C > 0$ that the next $N^3$ graphs are $W_{p(1)}, \ldots, W_{p(N^3)}$. We denote this event by $\Psi(t)$. In $t - 2D_E + 1 \leq \tau \leq t - D_E$, we have at least $\lfloor \frac{D_E}{N^3} \rfloor$ independent attempts to generate this sequence. Hence for some shift $1 \leq l_0 \leq N$

$$\mathbb{P}\left( \bigcap_{\tau=t-2D_E+1}^{t-D_E} \Psi^c(\tau) \right) \leq \mathbb{P}\left( \bigcap_{j=0}^{\lfloor \frac{D_E}{N^3} \rfloor - 1} \Psi^c\left( jN^3 + l_0 \right) \right) \leq (1 - p_C)^{\lfloor \frac{D_E}{N^3} \rfloor} \leq E^{-S_1 \kappa_0 \log E} \quad (63)$$

where $\kappa_0 > 0$ is a constant that depends on the distribution of $W(t)$.

Hence if $\Psi(\tau)$ occurred for some $t - 2D_E + 1 \leq \tau \leq t - D_E$ then thanks to positive diagonal of $W(t)$, $\prod_{\tau=t-2D_E+1}^{t-D_E} W(\tau)$ has only positive elements. This implies that a message from agent 1 will reach all other agents in this period of time, and messages from all other agents will reach agent 1 in this period of time. We can therefore obtain from the union bound that

$$1 - \mathbb{P}(S_C) \leq E \max_{t \in \mathcal{T}_e} \mathbb{P}\left( \bigcap_{\tau=t-2D_E+1}^{t-D_E} \Psi^c(\tau) \right) \leq E^{1 - S_1 \kappa_0 \log E}. \quad (64)$$

### 12.1.2 $\mathbb{E}\{b_{1,n}(t)\}$ is bounded from below by a positive constant

Let $U = \mathbb{E}\{W(t)\}$, which is fixed in time since $W(t)$ is i.i.d.. We have

$$\mathbb{E}\left\{ \prod_{\tau=t-2D_E+1}^{t-D_E} W(\tau) \right\} = \prod_{\tau=t-2D_E+1}^{t-D_E} \mathbb{E}\{W(\tau)\} = U^{D_E}. \quad (65)$$

Since $U = \mathbb{E}\{W(t)\}$ is a weight matrix of the connected graph $\bigcup_{i=1}^{M} G_i$, then $U^{q_0}$ has only positive elements for some integer $0 < q_0 \leq N \leq D_E$ (the diameter of this graph). Since $U$ is stochastic, so is $U^{D_E - q_0}$, and hence the minimal element of $U^{D_E} = U^{D_E - q_0} U^{q_0}$ is at least as large as the minimal element of $U^{q_0}$. Therefore, we conclude that for some positive constant $\kappa_1$ that depends on the distribution of $W(t)$:

$$\mathbb{E}\{b_{1,n}(t)\} = \mathbb{E}\left\{ \left[ \prod_{\tau=t-2D_E+1}^{t-D_E} W(\tau) \right]_{1,n} \right\} \geq \min_{i,j} U_{i,j}^{q_0} \triangleq \kappa_1 > 0. \quad (66)$$

### 12.1.3 Clean event for stochastic "arms" with memory

Define the filtration

$$\mathcal{F}_e = \sigma\left( \left\{ \boldsymbol{a}(t), \{z_n(t)\}_{n \in \mathcal{N}} \mid t < T_e \right\} \right) \quad (67)$$

so $R_n^e = R_n(T_e - 1)$ is $\mathcal{F}_e$-measurable. We define the effective reward at turn $t$ by

$$r(\boldsymbol{a}(t)) = \sum_{n=1}^{N} b_{1,n}(t + 2D_E - 1) R_n^e r_n(\boldsymbol{a}(t)). \quad (68)$$

Then, given $\boldsymbol{a}(t) = \boldsymbol{a}$, the expectation of $r(\boldsymbol{a})$ with respect to both $b_{1,n}(t + 2D_E - 1)$ and $r_n(\boldsymbol{a})$ is

$$\mu(\boldsymbol{a}) = \mathbb{E}\left\{ \sum_{n=1}^{N} b_{1,n}(t + 2D_E - 1) R_n^e r_n(\boldsymbol{a}) \mid \mathcal{F}_e \right\} \overset{(a)}{=} \sum_{n=1}^{N} R_n^e \mathbb{E}\{b_{1,n}(t + 2D_E - 1)\} \mu_n(\boldsymbol{a}) \quad (69)$$

where (a) uses that $z_n(t) = r_n(\boldsymbol{a}(t)) - \mu_n(\boldsymbol{a}(t))$ is independent of $b_{1,n}(t + 2D_E - 1)$, and both are independent of $\mathcal{F}_e$ since $b_{1,n}(t + 2D_E - 1)$ only depends on $G(t + D_E - 1), \ldots, G(t)$.

Hence, given that $\boldsymbol{a}(t) = \boldsymbol{a}$, $\mu(\boldsymbol{a}(t))$ and $\mathbb{E}\{b_{1,n}(t)\}$ are fixed in time since $W(t)$ is i.i.d. and $b_{1,n}(t) = \left[\prod_{\tau=t-2D_E+1}^{t-D_E} W(\tau)\right]_{1,n}$. We can now redefine the approximate max-weight action profile for epoch $e$ as:

$$\boldsymbol{a}_e^* = \arg\max_{\boldsymbol{a}} \sum_{n=1}^{N} \mathbb{E}\{b_{1,n}(t)\} R_n^e \mu_n(\boldsymbol{a}). \tag{70}$$

Let $r_{\max} = \max_n R_n^e$ and note that $r(\boldsymbol{a}(t)) \le r_{\max}$ for all $t \in \mathcal{T}_e$. As explained above, if the communication succeeded (i.e., $S_C$ occurred) in epoch $e$, then all agents know $r_{\max}$. For shorthand, denote the number of visits to action profile $\boldsymbol{a}$ up to time $t - 2D_E$ by $V = v_{\boldsymbol{a}}^e(t - 2D_E)$, and denote the corresponding turns by $t_1, \ldots, t_V$. The sequence of stochastic rewards $r(\boldsymbol{a}(t_1)), \ldots, r(\boldsymbol{a}(t_V))$ is not independent, since adjacent rewards depend on the same random communication graphs. While the visit turns $t_1, \ldots, t_V$ are random, in the worst case they are consecutive. Hence, we divide this sequence into $D_E$ groups: the first starts from $\tau = 1$ and proceeds with jumps of $D_E$ turns, the second starts from $\tau = 2$ and proceeds with jumps of $D_E$ turns and so on for $i = 1, \ldots, D_E$. We also denote the number of visits to $\boldsymbol{a}$ in group $i$ by $V_i$, and note that either $V_i = \left\lfloor \frac{V}{D_E} \right\rfloor$ or $V_i = \left\lfloor \frac{V}{D_E} \right\rfloor + 1$, such that $\sum_{i=1}^{D_E} V_i = V$. Then, each such subsequence is i.i.d. and we can apply Hoeffding's inequality on the subsequences separately regardless of the values of $t_1, \ldots, t_V$.

Therefore we have for every $\boldsymbol{a}$ that

$$\mathbb{P}\left(\left|\frac{1}{V}\sum_{\tau=1}^{V} r(\boldsymbol{a}(\tau)) - \mu(\boldsymbol{a})\right| \ge \rho_{\boldsymbol{a}}(t)\right) =$$

$$\mathbb{P}\left(\left|\frac{1}{V}\sum_{i=1}^{D_E}\sum_{j=0}^{V_i-1} r(\boldsymbol{a}(jD_E + i)) - \mu(\boldsymbol{a})\right| \ge \rho_{\boldsymbol{a}}(t)\right) \underset{(a)}{\le}$$

$$D_E \max_i \mathbb{P}\left(\left|\frac{1}{V_i}\sum_{j=0}^{V_i-1} r(\boldsymbol{a}(jD_E + i)) - \mu(\boldsymbol{a})\right| \ge \rho_{\boldsymbol{a}}(t)\right) \underset{(b)}{\le}$$

$$2D_E e^{-\frac{2\rho_{\boldsymbol{a}}^2(t)}{r_{\max}^2}\left\lfloor \frac{v_{\boldsymbol{a}}^e(t-2D_E)}{D_E} \right\rfloor} \underset{(c)}{\le} 2D_E E^{-2} \tag{71}$$

where (a) is the union bound, since for

$$\sum_{i=1}^{D_E}\sum_{j=0}^{V_i-1} r_n(\boldsymbol{a}(jD_E + i)) \ge V(\mu(\boldsymbol{a}) + \rho_{\boldsymbol{a}}(t)) \tag{72}$$

at least one of the $D_E$ inner sums needs to be at least $V_i(\rho_{\boldsymbol{a}}(t) + \mu(\boldsymbol{a}))$, with the same argument on

$$\sum_{i=1}^{D_E}\sum_{j=0}^{V_i-1} r_n(\boldsymbol{a}(jD_E + i)) < V(\mu(\boldsymbol{a}) - \rho_{\boldsymbol{a}}(t)). \tag{73}$$

Inequality (b) is then Hoeffding's inequality for i.i.d. bounded random variables, and (c) uses $\rho_{\boldsymbol{a}}(t) = \sqrt{\frac{2r_{\max}^2 D_E \log E}{\max\{v_{\boldsymbol{a}}^e(t-2D_E),1\}}}$.

### 12.1.4 Regret of learning an approximately optimal action profile

Define the regret of approximating $\boldsymbol{a}_e^*$ by

$$R_E \triangleq \sum_{t=T_e}^{T_e+E-1} \sum_{n=1}^{N} b_{1,n}(t + 2D_E) R_n^e (\mu_n(\boldsymbol{a}_e^*) - r_n(\boldsymbol{a}(t+1))). \tag{74}$$

Repeating the argument in (33) on the new confidence interval yields

$$\Delta_{\boldsymbol{a}} = \mu\left(\boldsymbol{a}_e^*\right) - \mu\left(\boldsymbol{a}\right) \le 4\sqrt{\frac{2r_{\max}^2 D_E \log E}{\max\left\{v_{\boldsymbol{a}}^e\left(T_E + E - 1\right) - 1 - n_{\boldsymbol{a}}^e\left(T_E + E - 1\right), 1\right\}}}$$

$$\implies v_{\boldsymbol{a}}^e\left(T_E + E - 1\right) - n_{\boldsymbol{a}}^e\left(T_E + E - 1\right) - 1 \le \frac{32r_{\max}^2 D_E \log E}{\Delta_{\boldsymbol{a}}^2}. \quad (75)$$

For $\mathcal{A}_e^*$ as defined in (26) but for the new $r_{\max}$, we now obtain

$$\mathbb{E}\left\{R_E \mid S_C \cap A_G, \mathcal{F}_e\right\} \underset{(a)}{\le}$$

$$2D_E S_E r_{\max} + r_{\max}\Delta_E E + \sum_{\boldsymbol{a} \in \left(\mathcal{A}_e^*\right)^c \cap \mathcal{S}_E} \left(\frac{32r_{\max}^2 D_E \log E}{\Delta_{\boldsymbol{a}}} + \Delta_{\boldsymbol{a}}\left(n_{\boldsymbol{a}}^e\left(T_E + E - 1\right) + 1\right)\right) \underset{(b)}{\le}$$

$$r_{\max}\left(S_E \frac{32 D_E \log E}{\Delta_E} + 2D_E\left(1 + S_E\right) + S_E + \Delta_E E\right) \underset{(c)}{\le}$$

$$\left(\left(64S_0 + 1\right)\log^2 E\sqrt{2S_1 E} + 14S_0 S_1 \log^3 E\right)\max_n R_n^e \quad (76)$$

where (a) uses (71) along with the union bound over the $S_E$ actions profiles and $E$ turns. Inequality (b) uses $\Delta_{\boldsymbol{a}} \ge \Delta_E r_{\max}$ for all $\boldsymbol{a} \in \left(\mathcal{A}_e^*\right)^c$, and also that $\Delta_{\boldsymbol{a}} \le r_{\max}$ and then that $\sum_{\boldsymbol{a} \in \left(\mathcal{A}_e^*\right)^c \cap \mathcal{S}_E} n_{\boldsymbol{a}}^e\left(T_E + E - 1\right) \le 2D_E$ which follows since we played exactly $2D_E$ action profiles between turn $T_E + E - 1 - 2D_E$ and turn $T_E + E - 1$, and in the worst case they are all in $\left(\mathcal{A}_e^*\right)^c$. Inequality (c) uses that $r_{\max} = \max_n R_n^e$, $\Delta_E = \log E\sqrt{\frac{D_E}{E}}$, $S_E \le 2S_0 \log E$ and $D_E \le 2S_1 \log^2 E$.

We conclude that

$$\mathbb{E}\left\{R_E \mid \mathcal{F}_e\right\} = \mathbb{E}\left\{\sum_{t=T_e}^{T_e + E - 1}\sum_{n=1}^{N} b_{1,n}\left(t + 2D_E\right) R_n^e\left(\mu_n\left(\boldsymbol{a}_e^*\right) - r_n\left(\boldsymbol{a}\left(t+1\right)\right)\right) \mid \mathcal{F}_e\right\} =$$

$$\mathbb{E}\left\{R_E \mid S_C \cap A_G, \mathcal{F}_e\right\}\mathbb{P}\left(S_C \cap A_G \mid \mathcal{F}_e\right) + \mathbb{E}\left\{R_E \mid S_C^c \cup A_G^c, \mathcal{F}_e\right\}\mathbb{P}\left(S_C^c \cup A_G^c \mid \mathcal{F}_e\right) \underset{(a)}{\le}$$

$$\left(\left(64S_0 + 1\right)\log^2 E\sqrt{2S_1 E} + 14S_0 S_1 \log^3 E\right)\max_n R_n^e +$$

$$\left(E^{2 - S_1 \kappa_0 \log E} + E^{1 - \frac{S_0|\mathcal{A}_e^*|}{KN}}\right)\max_n R_n^e \underset{(b)}{=} \kappa_1 \varepsilon_E E \max_n R_n^e \quad (77)$$

where in (a) we used that $\mathbb{E}\left\{R_E \mid S_C^c \cup A_G^c, \mathcal{F}_e\right\} \le E\max_n R_n^e$, the error bounds in (28) and (64), and (76). In equality (b) we used the definition of $\varepsilon_T$ in (60).

#### 12.1.5 Lyapunov Analysis

We repeat the proof of Theorem 2, with $b_{1,n}\left(t + 2D_E\right)$ replacing $b_{1,n}$, and detail the necessary modifications. Let $\lambda_{\min}, \lambda_{\max}$ be the minimal and maximal values of $\boldsymbol{\lambda}$, respectively. As before, the QoS regret of agent $n$ at turn $t$ cannot be too far from its QoS regret at the beginning of epoch $e = e\left(t\right)$:

$$R_n\left(t\right) \ge R_n^e - \sum_{\tau=T_e}^{t}\left(r_n\left(\boldsymbol{a}\left(\tau\right)\right) - \lambda_n\right) \ge R_n^e - \left(1 - \lambda_n\right)E_T, \quad (78)$$

so by summing over all turns in epoch $e$ and all agents, we obtain

$$\sum_{t=T_e}^{T_e+E_T-1} \sum_{n=1}^{N} \mathbb{E}\left\{b_{1,n}\left(t+2D_E\right)\right\} R_n\left(t\right) \geq$$

$$\sum_{t=T_e}^{T_e+E_T-1} \sum_{n=1}^{N} \mathbb{E}\left\{b_{1,n}\left(t+2D_E\right)\right\} R_n^e - E_T \sum_{t=T_e}^{T_e+E_T-1} \sum_{n=1}^{N} \mathbb{E}\left\{b_{1,n}\left(t+2D_E\right)\right\}\left(1-\lambda_n\right) \underset{(a)}{\geq}$$

$$\kappa_1 E_T \max_n R_n^e - \left(1-\lambda_{\min}\right) E_T^2 \quad (79)$$

where (a) uses that $\frac{\sum_{n=1}^{N} \mathbb{E}\{b_{1,n}(t)\} R_n^e}{\max_n R_n^e} \geq \kappa_1$ for the constant $\kappa_1 = \min_n \mathbb{E}\left\{b_{1,n}\left(t\right)\right\} > 0$.

We can now bound from below the expected weighted reward as follows:

$$\sum_{t=0}^{T-1} \sum_{n=1}^{N} \mathbb{E}\left\{b_{1,n}\left(t+2D_E\right) R_n\left(t\right) r_n\left(\boldsymbol{a}\left(t+1\right)\right)\right\} =$$

$$\sum_{t=0}^{T-1} \sum_{n=1}^{N} \mathbb{E}\left\{b_{1,n}\left(t+2D_E\right) r_n\left(\boldsymbol{a}\left(t+1\right)\right)\left(R_n^{e(t)} + R_n\left(t\right) - R_n^{e(t)}\right)\right\} \underset{(a)}{\geq}$$

$$\sum_{t=0}^{T-1} \sum_{n=1}^{N} \mathbb{E}\left\{b_{1,n}\left(t+2D_E\right)\right\} \mathbb{E}\left\{R_n^{e(t)} \mu_n\left(\boldsymbol{a}_{e(t)}^*\right)\right\}$$

$$- \left(1-\lambda_{\min}\right) E_T T - \kappa_1 \varepsilon_{E_T} E_T \sum_{e=1}^{\left\lceil \frac{T}{E_T} \right\rceil} \mathbb{E}\left\{\max_n R_n^e\right\} \underset{(b)}{\geq}$$

$$\sum_{t=0}^{T-1} \sum_{n=1}^{N} \mathbb{E}\left\{b_{1,n}\left(t+2D_E\right)\right\} \mathbb{E}\left\{R_n\left(t\right) \mu_n\left(\boldsymbol{a}^*\left(t+1\right)\right)\right\}$$

$$- \left(1+\lambda_{\max}-\lambda_{\min}\right) E_T T - \kappa_1 \varepsilon_{E_T} E_T \sum_{e=1}^{\left\lceil \frac{T}{E_T} \right\rceil} \mathbb{E}\left\{\max_n R_n^e\right\} \quad (80)$$

where (a) uses (78) and (77) with the tower rule on $\mathbb{E}\left\{\cdot \mid \mathcal{F}_{e(t)}\right\}$, so

$$\mathbb{E}\left\{\mathbb{E}\left\{b_{1,n}\left(t+2D_E\right) R_n^{e(t)} \mu_n\left(\boldsymbol{a}_{e(t)}^*\right) \mid \mathcal{F}_{e(t)}\right\}\right\} =$$

$$\mathbb{E}\left\{b_{1,n}\left(t+2D_E\right)\right\} \mathbb{E}\left\{R_n^{e(t)} \mu_n\left(\boldsymbol{a}_{e(t)}^*\right)\right\}. \quad (81)$$

In (a), we also used the definition of $\varepsilon_{E_T}$ in (60), along with the fact that $|\mathcal{A}_e^*| \geq \min_{\boldsymbol{\alpha} \in \Delta} |\mathcal{A}^*\left(\boldsymbol{\alpha}\right)|$ for all $e$. Inequality (b) uses Lemma 2 with $\mathbb{E}\left\{b_{1,n}\left(t+2D_E\right)\right\}$ replacing $b_{1,n}$.

Therefore, by using Lemma 1 with $b_{1,n}(t + 2D_E)$ replacing $b_{1,n}$ we obtain

$$
\mathbb{E}\left\{L(T) - L(0)\right\} = \sum_{t=0}^{T-1} \mathbb{E}\left\{L(t+1) - L(t)\right\} \underset{(a)}{\leq}
$$

$$
2T - 2\sum_{t=0}^{T-1}\sum_{n=1}^{N} \mathbb{E}\left\{b_{1,n}(t+2D_E)\right\} \mathbb{E}\left\{R_n(t)\mu_n(\boldsymbol{a}^*(t+1))\right\}
$$

$$
+ 2\sum_{t=0}^{T-1}\sum_{n=1}^{N} \lambda_n \mathbb{E}\left\{R_n(t) b_{1,n}(t+2D_E)\right\}
$$

$$
+ 2(1 + \lambda_{\max} - \lambda_{\min})E_T T + 2\kappa_1 \varepsilon_{E_T} E_T \sum_{e=1}^{\left\lceil \frac{T}{E_T} \right\rceil} \mathbb{E}\left\{\max_n R_n^e\right\} \underset{(b)}{\leq}
$$

$$
2T - 2(\delta(\boldsymbol{\lambda}, E_T) + \varepsilon_{E_T}) \sum_{t=0}^{T-1}\sum_{n=1}^{N} \mathbb{E}\left\{R_n(t)\right\} \mathbb{E}\left\{b_{1,n}(t+2D_E)\right\}
$$

$$
+ 2(1 + \lambda_{\max} - \lambda_{\min})E_T T + 2\kappa_1 \varepsilon_{E_T} E_T \sum_{e=1}^{\left\lceil \frac{T}{E_T} \right\rceil} \mathbb{E}\left\{\max_n R_n^e\right\} \underset{(c)}{\leq}
$$

$$
2T - 2\delta(\boldsymbol{\lambda}, E_T) \sum_{t=0}^{T-1}\sum_{n=1}^{N} \mathbb{E}\left\{R_n(t)\right\} \mathbb{E}\left\{b_{1,n}(t+2D_E)\right\}
$$

$$
+ 2(1 + \lambda_{\max} - \lambda_{\min})E_T T + 2\varepsilon_{E_T}(1 - \lambda_{\min})E_T T \quad (82)
$$

where (a) uses Lemma 1 and (80). Inequality (b) uses that if $\boldsymbol{\lambda} \in C_{\varepsilon_{E_T}}(\mathcal{G})$, then

$$
\sum_{n=1}^{N} \mathbb{E}\left\{b_{1,n}(t+2D_E)\right\} R_n(t) \mathbb{E}\left\{\mu_n(\boldsymbol{a}^*(t+1)) \mid \mathcal{F}(t)\right\} \geq
$$

$$
\sum_{n=1}^{N} \mathbb{E}\left\{b_{1,n}(t+2D_E)\right\} R_n(t) \mathbb{E}^{\boldsymbol{P}}\left\{\mu_n(\boldsymbol{a})\right\} \geq
$$

$$
\sum_{n=1}^{N} \mathbb{E}\left\{b_{1,n}(t+2D_E)\right\} R_n(t) (\lambda_n + \delta(\boldsymbol{\lambda}, E_T) + \varepsilon_{E_T}) \quad (83)
$$

and also that
$$
\mathbb{E}\left\{R_n(t) b_{1,n}(t+2D_E)\right\} = \mathbb{E}\left\{R_n(t)\right\} \mathbb{E}\left\{b_{1,n}(t+2D_E)\right\} \quad (84)
$$
since $R_n(t)$ and $b_{1,n}(t+2D_E)$ are independent: $R_n(t)$ is $\mathcal{F}(t)$-measurable, while $b_{1,n}(t+2D_E)$ depends on $G(t+D_E), \ldots, G(t+1)$ that were generated after the action profile $\boldsymbol{a}(t)$ was played. Finally, inequality (c) in (82) uses (79).

Since $L(0) = 0$ and $L(T) \geq 0$, we conclude from (82) that

$$
2\delta(\boldsymbol{\lambda}, E_T) \sum_{t=0}^{T-1}\sum_{n=1}^{N} \mathbb{E}\left\{R_n(t)\right\} \mathbb{E}\left\{b_{1,n}(t+2D_E)\right\} \leq
$$

$$
2T + 2(1 + \lambda_{\max} - \lambda_{\min})E_T T + 2\varepsilon_{E_T}(1 - \lambda_{\min})E_T T \quad (85)
$$

so if $(\lambda_1, \ldots, \lambda_N) \in C_{\varepsilon_{E_T}}(\mathcal{G})$ then

$$
\frac{1}{T}\sum_{t=0}^{T-1}\sum_{n=1}^{N} \mathbb{E}\left\{b_{1,n}\right\} \mathbb{E}\left\{R_n(t)\right\} \leq \frac{1 + (2 + \varepsilon_{E_T})E_T}{\delta(\boldsymbol{\lambda}, E_T)} \quad (86)
$$

and the rest of the proof follows like that of Theorem 2 since we showed that $\mathbb{E}\left\{b_{1,n}\right\} \geq \kappa_1 > 0$ for all $n$, and $\sum_{n=1}^{N} b_{1,n}(t) = 1$ with probability 1.