# OpenReview forum: "Queue Up Your Regrets: Achieving the Dynamic Capacity Region of Multiplayer Bandits"
_NeurIPS.cc/2022/Conference — NeurIPS 2022 Accept_

### Official Review · Reviewer_2EXT · 2022-07-09

**Rating:** 7
**Confidence:** 3
**Soundness:** 3 good
**Presentation:** 3 good
**Contribution:** 2 fair

**Summary:**

The paper examines the problem of multi-agent bandits (with given initial actions) from a distributional (i.e. decentralized) point of view, in which the actions result in an optimal matching to agnostic agents.
The formulation belongs to the area of optimization using a game theory setting
and, in their randomized technique, the authors suggest in each round the selection of a subset of available strategies.
The invariant of their algorithm in every iteration is the maximization of the expected (weighted) utility.
For the complexity of the algorithm, there is an upper bound in the expected utility which suffices as a guarantee in the chosen count of shared messages (O(communication graph's diameter)).

**Questions:**

Some questions were posed in the 'Weaknesses' section. Also:
1. Is the set of actions common for all players (e.g. lines 90 and 191)? Therefore, is the difference located in the different rewards?
2. Is the right hand side of (13) of O(communication graph's diameter)?
3. How is parameter 'E' selected before (and during) Algorithm 1?
4. Could the authors make a more direct comment on the individual accumulated Regret R(t) 'behaving like a queue' and the stability? I am not sure there is a theorem of queuing theory fashion here.
5. Is the set of sets of approximately optimal action profiles reducing to a particular complexity class of games?


**Limitations:**

The paper has a theoretical character and aspires to suggest a complexity result through an optimization model. Given that this is a conference about Learning and related fields,
according to my opinion,
the authors are encouraged to involve a comment and designate how the optimization iterations of their algorithm track a learning course/task (e.g. based on some dataset) both in their model and in the 2 numerical simulations of this process.

POST REBUTTAL: After reviewing the authors' response to all reviewers and after the discussion with the other reviewers, I decided to increase my score to 'Accept'. It seems that the authors submitted an improved version of the Supplementary Material, with enhanced parts on the example of graphs with reference to time evolution. It seems that their effort used to create a novel work is reflected on their paper and answers.


**Strengths And Weaknesses:**

Strengths: The formulation involving the suggested algorithm seems to be a novel model; the paper employs a method which doesn't seem to be already explored in this multiplayer setting for the purpose of achieving a feasible Quality of Service with a sublinear expected regret.

There are arithmetic examples of the model which demonstrate and strengthen the applicability of its theory.

It describes the problem of the reward maximization both with an intuitive and a rigorous way, which allows the reader to follow the developed model.

It seems technically sound, it involves a scalability part which makes it more realistic than a theoretical construction and it has the interesting idea/trick of the search to a limited random subset of action profiles. From a first reading, there doesn't seem to be any missing step or error in the proofs.

Weaknesses: 1. Learning fashion could be discussed. The examples focus in the convergence of the algorithm to \lambda. Is that the (machine) 'learning' context of the model? For example, would there be any setting using data (instead of random initial values) for the \lambda inference or the learning of any other parameter?
It seems that the assumptions of p_n_k in 5.1 'help' the example to validate the model (although this is in enough as a sanity check for a new model; yet it also seems to restrict it).

A possible interpretation is that the model belongs to Unsupervised Learning; in each epoch the regret is assessed (not in a gradient based way).

2. Impact could be discussed. Could the authors elaborate on the impact of this model in comparison to existing models of bandits. The paper lacks any kind of comparisons to existing models, e.g. about the complexity of its algorithm, the convergence speed, any other metrics that exhibit superiority. Are there any such justifications of better results that make this algorithm needed? How is this model guaranteeing stability  of the QoS regrets differently from other algorithms?

---

> ### Author Response · Authors · 2022-08-02
> **Response to Reviewer #3: Part 1**
>
> We thank the reviewer for the positive evaluation and the detailed feedback that helps to improve the paper. We hope that our response below addresses the issues you raised.
>
> **"Weaknesses: 1. Learning fashion could be discussed..."**
>
> We wholeheartedly agree with the reviewer that the learning context and the practical scenarios to which the algorithm applies are very important. We would like to clarify that $\lambda_{n}$ is an input parameter that dictates the desired Quality of Service (e.g., throughput) that we want to guarantee for player n. This is typically chosen by the network operator (wireless networks) or the owner of the robots. Therefore the goal is not to estimate this parameter (or any other parameter).
>
> The assumptions on $p\_{n}^{k}$ are actually a classical packet transmission model in wireless networks [Tse & Viswanath 2005], which is successful mostly due to its reliability in practice. In fact, our work allows us to use this realistic model rather than the less practical ones considered in previous work that cannot model the geometry of the devices. In practice, it is the locations of the devices that determine the channel gains. Hence, by randomizing the channel gains, we show that our algorithm works for many possible scenarios. A random location for a wireless device is indeed the case in practice since smartphones, laptops, and access points are placed by the users. Hence, for wireless networks, generating data synthetically by randomizing the locations and using the channel gains model is a good approximation of real data.
>
> Another motivating example is that of cloud computing and congestion games. Consider N edge devices that request computation time from the same server. In every given turn, each device can either access the server or not. If $m$ devices access the server, each gets a reward (CPU time) of $\frac{1}{m^{p}}$ for some $p\geq1$. This models the overhead that the server incurs by having to run more than one job at a time, where p=1 represents no overhead. The overhead can result from context switches, limited RAM, and communication with the devices. Using our approach, the devices can learn efficient time-sharing as opposed to simultaneously accessing the server. For $p=2$ and two devices, a simple time-sharing can achieve $\lambda\_{1}=\lambda\_{2}=\frac{1}{2}$ as opposed to $\lambda\_{1}=\lambda\_{2}=\frac{1}{4}$ with simultaneous access. This scenario is a special case of a congestion game that can easily be extended to K servers (resources), and different job sizes $w\_{n}$ for each device.
>
> **"A possible interpretation is that the model belongs to Unsupervised Learning..."**
>
> Our work falls under the general framework of online learning or reinforcement learning, where players act online and receive a reward each turn, from an unknown model.
>
> **"Impact could be discussed..."**
>
> We are happy to say that currently there are no other algorithms that guarantee the stability of QoS regrets. This is why we believe that the contribution of our paper is significant. We refer the reviewer to lines 69-77 that compare our work to the multiplayer bandit literature. We would be happy to extend this discussion. The main difference can be illustrated using a simple example. Consider two players and one resource, such that the players cannot use the resource at the same time, so it results in zero reward for each. If one player gets the resource and the other does not, then that player gets a reward of one. A traditional multiplayer bandit algorithm that aims to maximize the sum of rewards would achieve a sum of rewards of one, and a minimal reward of zero, in all turns (by giving the resource to the same player all the time). Our algorithm would learn to give the resource to player 1 half of the time, and to player 2 for the other half. Hence, over time, the sum of rewards would still be one, but the minimal reward is now 0.5. This larger capacity region is the main advantage of our algorithm over the existing multiplayer bandit literature. This example is given in lines 98-106.
>
> The computational complexity of our algorithm for player $n$ is $O\left(S_{E}+\nu_{n}\right)$, where $S_{E}$ is the size of the random subset of action profiles and $\nu_{n}$ is the degree of player n in the communication graph. This follows since in the worst case player n goes through a list of size $S_{E}$, and adds $\nu_{n}$ numbers received from its neighbors. Therefore the computational complexity is very modest. Following the reviewer's suggestion, we have added this discussion to the paper.
>
> The convergence speed is essentially given by the QoS regret bound in equation (7) which quantifies how much our algorithm loses compared to a perfect maximum weight matching that knows all of the problem parameters. Note that our algorithm does not converge to any stationary solution but learns a pattern of action profiles to jump between.

---

> > ### Author Response · Authors · 2022-08-02
> > **Response to Reviewer #3: Part 2**
> >
> > **"Is the set of actions common for all players (e.g. lines 90 and 191)? Therefore, is the difference located in the different rewards?"**
> >
> > Yes, the set of actions is the same for all players. Of course, every player can pick a different action.
> >
> > **"Is the right hand side of (13) of O(communication graph's diameter)?"**
> >
> > The RHS of (13) would be the same regardless of the communication overhead since this would be the regret of a single agent trying to maximize $\sum\_{n=1}^{N}b_{1,n}R\_{n}^{e}$. Our algorithm aims to minimize the communication required to implement this, which is indeed $O\left(diameter\right)$.
> >
> > **"How is parameter 'E' selected before (and during) Algorithm 1?"**
> >
> > $E$ is chosen only before the algorithm and is kept constant. Choosing a large E would enlarge the region of $\boldsymbol{\lambda}$ for which the algorithm would work, and choosing a small E will minimize the QoS regret (delay). Using $E\_{T}$ that increases with $T$ (even $E\_{T}=\log\left(\log T\right)$) will guarantee that for large $T$, the whole interior of the capacity region is feasible.
> >
> > **"Could the authors make a more direct comment on the individual accumulated Regret R(t) 'behaving like a queue' and the stability? I am not sure there is a theorem of queuing theory fashion here"**
> >
> > The observation that the QoS regret behaves like a queue is what inspired our algorithm. In this analogy, a “packet” arrives with rate $\lambda_{n}$, and when a player gets a reward of $r_{n}\left(\boldsymbol{a}\right)$, it is like a departure rate of $r_{n}\left(\boldsymbol{a}\right)$ since $r_{n}\left(\boldsymbol{a}\right)$ units will leave the queue. Hence the QoS regret $R\_{n}\left(t\right)=\max[0,\lambda\_{n}t-\sum\_{\tau=1}^{t}r\_{n}\left(\boldsymbol{a}\left(\tau\right)\right)]$ tracks how many “packets” are still in the queue. Our algorithm is a distributed version of maximum-weight matching for this “weird queue”. Maximum-weight matching is a key algorithm in queuing theory, and our analysis is inspired by that of maximum-weight matching while dealing with the subtleties that arise from this non-standard queue.
> >
> > **"Is the set of sets of approximately optimal action profiles reducing to a particular complexity class of games?"**
> >
> > Yes. For example, imagine that we have $K<N$ resources, such that every player gets a reward of one by using a resource alone, but if two or more players use the same resource they all get zero reward. Then maximizing $\sum\_{n}b_{1,n}R\_{n}r\_{n}\left(\boldsymbol{a}\right)$ amounts to sorting the players according to their weight $b_{1,n}R_{n}$ and letting the players with the K-maximal weights use the resources. There are $K!\left(N-K\right)!$ such optimal action profiles, so the set of approximately optimal action profiles is large and easy to compute.
> >
> > **Limitations**
> >
> > The iterations of our algorithm correspond to the real-time decision-making turns of the agents. Hence, in terms of learning, our work falls under the general notion of “multi-agent reinforcement learning” where agents make decisions in real-time in an unknown environment. In Section 5, we gave two practical numerical examples, which in our view give the “data” aspect (see the discussion above regarding wireless networks, and the new example for cloud computing). Other than that the approach is not data-driven, and the learning is reinforcement (reward) based.

---

### Official Review · Reviewer_NWYo · 2022-07-10

**Rating:** 4
**Confidence:** 5
**Soundness:** 2 fair
**Presentation:** 2 fair
**Contribution:** 2 fair

**Summary:**

Let $L(a, n) := \lambda_n - r_n(a)$.
This paper consider the minimax problem $p = \arg\min_{p \in \Delta(a)} \max_n L(a, n)$ (This equivalence holds when the regret is non-negative).
The authors propose an algorithm to solve the above problem under the distributed setting, where $a = (a_1, \ldots, a_N)$ is chosen by $N$ cooperative agents.
Particularly, $p$ is approximated through $T$ deterministic distribution, i.e., $p \approx \frac{1}{T}\sum_{t=1}^T \delta_{a^t}$.

**Questions:**

The powerful tools in the literature of matrix game are missed, which can help improve the results greatly.

**Limitations:**

The powerful tools in the literature of matrix game are missed, which can help improve the results greatly.

**Strengths And Weaknesses:**

The above mentioned minimax problem is generally known as matrix game, which can be solved efficiently through the EXP3 algorithm with full observations, and the EXP3-IX algorithm with partial observations (see the book "Bandit Algorithms" for details).
The above two algorithms both have the regret bound $\widetilde{O}(\sqrt{K^NT})$, which is much better than Theorem 2, and holds under a broader condition when $\lambda \in C(\mathcal{G})$.
The authors should incorporate the missing powerful tools into the distributed setting to improve their results.
Moreover, the OMWU and PU algorithms in the literature of matrix game may help improve the $\sqrt{T}$-regret to $\log(T)$-regret.

The authors may also discuss the possibility to avoiding the exponential dependence on $N$ based on the potential game or other models.

---

> ### Author Response · Authors · 2022-08-02
> **Response to Reviewer #2: Part 1**
>
> We thank the reviewer for this insightful view on the problem. We believe that it can be used to better motivate our maximum-weighted matching approach. The reason is that it highlights some subtleties that can help to illustrate why our QoS regret is unique compared to the online learning literature. Our detailed response below consists of two parts:
>
> - In part 1 we explain how our Theorem 2 provides stronger performance guarantees than just bounding the expected QoS regret at turn t. It bounds the average expected QoS regret over time by a constant with respect to $T$.
>
> - In part 2 we explain that while at first glance it looks reasonable and even tempting to tackle our problem as a zero-sum matrix game, this would not have expected QoS regret guarantees and therefore cannot compete with our approach.
>
> We hope that the reviewer would appreciate these subtle technical details and our algorithmic advantages. In this case, we would appreciate it if the reviewer can reevaluate our paper based on this more detailed discussion.
>
> Part 1:
>
> First, we would like to spotlight some crucial points regarding our main result:
>
> 1. The QoS expected reward we prove is indeed $\tilde{O}\left(\sqrt{T}\right)$. It follows since $E_{T}$ can increase as $\log T$, or arbitrarily slower. In fact, we can choose a constant E as a function of $\boldsymbol{\lambda}$, so the E factor becomes better for “easy problems” (where $\boldsymbol{\lambda}$ is deep inside the interior of the capacity region). This property improves the proposed $K^{N}$ factor significantly, allowing for adaptation to the problem at hand. Therefore, using our maximum-weight matching approach, one can improve the exponential dependence on $N$ even in a general game, without having to assume a specific family of games like potential games.
>
> 2. The most important point we want to make here is that the performance guarantee is much stronger than just proving $\tilde{O}\left(\sqrt{T}\right)$ QoS regret, which is but a corollary of our main result. The key is the bound in (7) in Theorem 2, which shows that the average expected reward over T turns is constant in $T$, $O\left(1\right)$. The $\tilde{O}\left(\sqrt{T}\right)$ bound just means that no QoS expected regret can exceed $\tilde{O}\left(\sqrt{T}\right)$, from $t=1$ to $t=T$. If there are even a few $\tilde{O}\left(\sqrt{T}\right)$ regrets in the sum in (7), it means that most of the expected QoS regrets are near zero. No min-max approach can achieve such a strong guarantee. We have now revised Theorem 1 to include this guarantee.
>
> 3. The average constant expected QoS regret (averaged over time from $t=1$ to $t=T$) is possible using our algorithm by exploiting the fact that the QoS regret is non-standard. The expected QoS Regret $E[R_{n}\left(t+1\right)]$  (or its bound) can be smaller than $E[R_{n}\left(t\right)]$  (or its bound). This is not the case with the standard regret in online learning. In fact, the QoS regret of the players will fluctuate such that when the QoS regret of a player becomes larger enough compared to others, the algorithm will tend to prioritize this player until her QoS regret will drop again. This cannot happen in an algorithm that does not take the QoS regret as input in an online manner (like that suggested by the reviewer).
>
> 4. Note that if $\boldsymbol{\lambda}\notin C\left(\mathcal{G}\right)$ then there exists a player n such that $\underset{p\in\Delta\left(\boldsymbol{a}\right)}{\arg\min}\left(\lambda_{n}-E^{\boldsymbol{a}\sim p}[r_{n}\left(\boldsymbol{a}\right)]\right)>0$, which means that the QoS regret of this player is linear in $T$ (i.e., trivial regret). Hence $\boldsymbol{\lambda}\in C\left(\mathcal{G}\right)$ is a necessary condition to have sublinear QoS regret for all players.

---

> > ### Author Response · Authors · 2022-08-02
> > **Response to Reviewer #2: Part 2**
> >
> > Part 2:
> >
> > 1. We have given some serious thought to the idea proposed by the reviewer, including a literature review about learning in matrix zero-sum games [Freund & Schapire 1997, Bailey & Piliouras 2018, Bailey 2021]. Our understanding is that the approach proposed by the reviewer can be used to estimate a “stabilizing distribution” $p$ such that $E^{\boldsymbol{a}\sim p}[r_{n}\left(\boldsymbol{a}\right)]\geq\lambda_{n}$ for all $n$. However, this approach is inferior to our approach for the following reasons:
> >
> > - Following the literature, the proposed approach shows that $\boldsymbol{p}\_{T}$ is an $\varepsilon_{T}$-Nash of the zero-sum game, where $\varepsilon_{T}\rightarrow0$ as $T\rightarrow\infty$. Indeed, all Nash equilibria of the zero-sum game are min-max strategies. However, since convergence to the set of NE is in the $\varepsilon_{T}$-Nash sense, this does not imply that $\left\Vert \boldsymbol{p}\_{T}-\boldsymbol{p}\_{\infty}\right\Vert\_{2}=O(\frac{1}{\sqrt{T}})$, where $\boldsymbol{p}\_{\infty}$ satisfies $E^{\boldsymbol{a}\sim\boldsymbol{p}\_{\infty}}[r_{n}\left(\boldsymbol{a}\right)]\geq\lambda\_{n}$ for all $n$. The missing link is to show how far away (in the Euclidean sense) an $\varepsilon_{T}$-Nash is from the set of exact Nash, which does not have to be $\varepsilon_{T}$, and can potentially be any function of $\varepsilon_{T}$. Hence, it is unclear how fast $\boldsymbol{p}\_{T}$ converges to $\boldsymbol{p}\_{\infty}$ in the Euclidean sense. Therefore it is unclear what expected regret $\boldsymbol{p}_{T}$ would have if used as a fixed strategy for $T$ turns.
> >
> > - Let us assume that $\boldsymbol{p}\_T$, once used, indeed accumulates an expected QoS regret of $O(\sqrt{T})$ for each player. This assumption is made for the sake of the argument, ignoring the difficulty we explained right above. Even then, the QoS regret of learning this $\boldsymbol{p}\_T$ is not $O\left(\sqrt{T}\right)$. To run the proposed algorithm, one agent will serve as the leader that picks $\boldsymbol{a}$ and communicates her choice to others. Additionally, this leader will play a zero-sum game against herself, also picking $n\_{t}$ every turn. Hence, this leader player implements a row player that picks $\boldsymbol{a}\_{t}$ to minimize $\lambda_{n\_{t}}-r\_{n\_{t}}\left(\boldsymbol{a}\_{t}\right)$ and a column player that picks $n\_{t}$ to maximize $\lambda\_{n\_{t}}-r\_{n\_{t}}\left(\boldsymbol{a}\_{t}\right)$. Using EXP3, the row player that picks $\boldsymbol{a}\_{t}$ will guarantee
> >
> > \begin{equation}
> > E[\sum\_{t=1}^{T}\left(\lambda\_{n\_{t}}-r\_{n\_{t}}\left(\boldsymbol{a}\_{t}\right)\right)-\underset{\boldsymbol{a}}{\min}\left(\sum\_{t=1}^{T}\left(\lambda_{n\_{t}}-r\_{n\_{t}}\left(\boldsymbol{a}\right)\right)\right)]=E[\underset{\boldsymbol{a}}{\max}\sum\_{t=1}^{T}r\_{n\_{t}}\left(\boldsymbol{a}\right)-\sum\_{t=1}^{T}r\_{n\_{t}}\left(\boldsymbol{a}\_{t}\right)]=O\left(\sqrt{T}\right)
> > \end{equation}
> >
> > and the column player that picks $n_{t}$ will guarantee
> >
> > \begin{equation}
> > E[\underset{n}{\max}\left(\sum\_{t=1}^{T}\left(\lambda\_{n}-r\_{n}\left(\boldsymbol{a}\_{t}\right)\right)\right)-\sum\_{t=1}^{T}\left(\lambda\_{n\_{t}}-r\_{n\_{t}}\left(\boldsymbol{a}\_{t}\right)\right)]=O\left(\sqrt{T}\right)
> > \end{equation}
> >
> > neither of these bound the QoS regret of any player $n$, which is $\sum\_{t=1}^{T}\left(\lambda\_{n}-r\_{n}\left(\boldsymbol{a}\_{t}\right)\right)$ (ignoring the $\max[0,x]$ operation), which in this abstraction corresponds to the sum of rewards of the EXP3 learner, and not to the regret.
> >
> > - The above regret guarantees can be used to show that
> >
> > \begin{equation}
> > E[\left|\sum\_{t=1}^{T}\left(\lambda\_{n\_{t}}-r\_{n\_{t}}\left(\boldsymbol{a}\_{t}\right)\right)-\underset{\boldsymbol{a}}{\min}\underset{n}{\max}\sum\_{t=1}^{T}\left(\lambda\_{n}-r\_{n}\left(\boldsymbol{a}\right)\right)\right|]=O(\sqrt{T})
> > \end{equation}
> >
> > which for $\boldsymbol{\lambda}$ in the capacity region also implies
> >
> > \begin{equation}
> > E[\sum_{t=1}^{T}\left(\lambda_{n_{t}}-r_{n_{t}}\left(\boldsymbol{a}_{t}\right)\right)]=O(\sqrt{T})
> > \end{equation}
> >
> > but this is not the QoS regret of any of the players, but a mix of part of the QoS regret of different players.
> >
> > We conclude that we cannot obtain any expected QoS regret guarantees using the proposed matrix game approach.
> >
> > 2. Even if some “leader-follower” online learning algorithm exists that does have expected QoS regret guarantees, this prospective algorithm would still be inferior to our proposed algorithm for two reasons:
> >
> > - As long as this algorithm does not have a constant (in $T$) expected QoS regret, it would not satisfy the bound in (7) in Theorem 2 that shows the constant average expected regret of our algorithm.
> >
> > - To communicate $\boldsymbol{a}$ to other players over a communication graph $G$ with diameter $d\left(G\right)$, a communication overhead of $O\left(Nd\left(G\right)\right)$ is needed. This is $N$ times worse than our algorithm requires.

---

### Official Review · Reviewer_1eJN · 2022-07-15

**Rating:** 7
**Confidence:** 4
**Soundness:** 3 good
**Presentation:** 2 fair
**Contribution:** 3 good

**Summary:**

The paper considers a system of $N$ cooperative agents who are trying to maximize their rewards in an online learning setting lasting $T$ time steps. Every agent's reward depends on the actions taken by all agents and not merely its own action, resulting in a multi-player MAB setting. Agents are allowed to communicate only over a fixed communication graph

The paper proposes an algorithm that attempts to mimic the celebrated MaxWeight algorithm, albeit in a decentralized manner, to ensure sublinear regret.

**Questions:**


1. The term "stability" needs to be defined in Sec. 2. The authors talk about stabilizing the regret queues without formally defining the term. Can you bring the definitions of strong stability and mean rate stability to Sec. 2?

2. Regarding the definition of the capacity region in Eqn. 2, I'm a bit concerned about the fact that $C_{\epsilon}$ doesn't seem to depend on the interference graph.
- Please correct the LHS of the definition to $\sum_a p(\mathbf{a})$ instead of $\sum_a \mathbf{p}(\mathbf{a})$. $\mathbf{p}$ is a vector, $p(\mathbf{a})$ is a scalar.
- Also, (I might be mistaken here) shouldn't the RHS of the inequality be "$\geq \lambda_n-\epsilon$" instead of "$> \lambda + \epsilon$"?

3. On line 132, shouldn't that be $r_n(\mathbf{a}(t))$?

4. Please refer to Lemma 2 somewhere in Lines 138-140.

5. I'm still not very clear about the tradeoffs that $E_T$ brings in. Regret scales with $\sqrt{TE_T}$, so it needs to grow strictly slower than  $T,$ but what lower bounds its rate of growth (I'm guessing delay)?

6. Communication graphs can change with time. T&E have a modified MaxWeight protocol for this as well, called "Serve the Longest Connected Queue." Is there any possibility of leveraging this protocol to extend your analysis to include a time varying topology?

**Limitations:**

The work is theoretical and I don't see any potential negative societal impact.

**Strengths And Weaknesses:**

Strengths:
1. The paper is well motivated, although concrete motivation is provided only in Sec. 5. I strongly recommend that the authors improve the explanation above Sec. 1.1 and situate the problem better.
2. I like the approach taken by the authors. MWM has been a mainstay in networking and its usage in solving multi-player MAB algorithms is very interesting.
3. The paper nicely follows Tassiulas and Ephremides's article, providing a capacity region, a (distributed) algorithm and stability guarantees. In my opinion, the discussion seems well rounded.

Weaknesses:
1. While the authors' attempt at introducing queueing and MW-related concepts to the machine learning community is laudable, I personally don't find much originality in either the algorithm or its analysis.

Consensus algorithms and approximate MWM have been proposed and analyzed in various scenarios in the past. The present algorithm doesn't appear to be more than a rehash of standard decentralized MWM with standard action elimination.

2. In fact, even the Lyapunov function is not non standard. Eqn (8) is the usual weighted 2-norm of the system backlog, and the proof of Lemma 8 (on pp 14) doesn't appear to contain anything surprising.

---

> ### Author Response · Authors · 2022-08-02
> **Response to Reviewer #1: Part 1**
>
> We thank the reviewer for the meticulous reading and the positive and constructive feedback. We hope that our response below resolves your concerns.
>
> **"I strongly recommend that the authors improve the explanation above Sec. 1.1 and situate the problem better"**
>
> This is a great idea. Thanks for pointing this out. We now bring some of the concrete motivation of Sec 5 to the introduction in the following (extended) paragraphs:
>
> - *“The QoS is often measured over time. Throughput in communication networks is only averaged over seconds, and progress in an automated factory is only measured over minutes. When resources are involved, time-sharing is sometimes better than simultaneous access. The optimal policy is often more complicated. For example, consider four agents and one resource, such that each agent can access the resource or not. If Agents 1 and 2 access the resource together, they each get zero reward, and the same is true for Agents 3 and 4. Otherwise, an agent gets a reward of 1 by accessing the resource. This simple example arises in many resource allocation applications. In channel allocation in wireless networks, far away devices can transmit simultaneously on the same channel, but nearby devices will have to pick different channels. In cloud computing, a job that requires little RAM can run simultaneously with a job that does not, but two RAM-intensive jobs cannot run share the CPU simultaneously. In electric vehicle charging, there is a limit to how many vehicles that share a transformer can charge simultaneously*.
>
> - *To achieve any feasible QoS, the agents need to learn when to access, but also, who to access with. The main objective of this paper is to design a distributed multi-agent algorithm that can achieve any (dynamically) feasible QoS based only on bandit feedback and minimal communication. Our algorithm can learn dynamic strategies such as time-sharing between groups of agents, therefore achieving QoS vectors that static multi-agent algorithms cannot. “*
>
> - Another motivating example is that of cloud computing and congestion games. Consider $N$ edge devices that request computation time from the same server. In every given turn, each device can either access the server or not. If $m$ devices access the server, each gets a reward (CPU time) of $\frac{1}{m^{p}}$ for some $p\geq1$. This models the overhead that the server incurs by having to run more than one job at a time, where $p=1$ represents no overhead. The overhead can result from context switches, limited RAM, and communication with the devices. Using our approach, the devices can learn efficient time-sharing as opposed to simultaneously accessing the server. For $p=2$ and two devices, a simple time-sharing can achieve $\lambda_{1}=\lambda_{2}=\frac{1}{2}$ as opposed to $\lambda_{1}=\lambda_{2}=\frac{1}{4}$ with simultaneous access. This scenario is a special case of a congestion game that can easily be extended to $K$ servers (resources), and different job sizes $w_{n}$ for each device.
>
> **Regarding Originality:**
>
> We believe that our paper has conceptual, structural, and technical originality, as detailed below (continued in the next comment):
>
> 1. The conceptual originality of our work is revealing and leveraging the analogy between Quality of Service (QoS) in multiagent learning, and queuing theory. This leads to using maximum-weight matching (MWM), which is a highly non-standard algorithm for multiagent learning. This analogy between QoS for multiagent scenarios and MWM was never considered before in the literature. From a queuing point of view, we agree that maximum-weight matching is indeed a standard algorithm, but from the same queuing point of view, the QoS regret is a highly non-standard queue, which leads to our structural originality detailed next.
>
> 2. The structural originality stems from the fact that our scenario is not a queuing scenario - there are no packets or transmissions. A standard queue is well defined by the backlog - no more information is needed. In contrast, our queue has “memory of being negative”: even if $R_{n}\left(t\right)=0$, we need to track $\lambda_{n}t-\sum_{\tau=1}^{t}r_{n}\left(\boldsymbol{a}\left(\tau\right)\right)$ which can be very negative, keeping $R_{n}\left(t\right)=0$ for a long time. This fact improves the scalability of our algorithm, as discussed in Section 7 of the appendix. In addition, the departure rate is an unknown function of the action of other agents and not of a single server, which necessitates a distributed learning algorithm to replace the traditional MWM. Moreover, our backlog is not an integer, and the arrival process is deterministic. Then, while a sublinearly increasing queue (in $T$) is unstable, in our case, as long as $\frac{R_{n}\left(T\right)}{T}\rightarrow0$, agent $n$ will asymptotically achieve its desired QoS. These subtleties are summarized in lines 216-225, which we edited to better explain how $R_{n}(t)$ has “memory of being negative”.

---

> > ### Author Response · Authors · 2022-08-02
> > **Response to Reviewer #1: Part 2**
> >
> > We also believe that our decentralized MWM is not standard as an algorithm. The combination of MWM, consensus, and action elimination has not been used before, and it has some constructive superpositions:
> >
> > - While consensus is often used for distributed optimization, the traditional consensus has to converge to the average agent value in the network, with a small error (which is never zero). This requires a doubly stochastic communication matrix. This is not the case with MWM, where converging to any linear combination of the initial agent values is good. For this reason, our consensus does not require a doubly stochastic matrix and converges with zero error after a finite number of iterations. We are not aware of any work that combined this enhanced consensus with MWM.
> >
> > - Successive elimination, as opposed to UCB for example, is much easier to distribute. The reason is that players do not need to know what other players are playing to deduce the index of the action profile. Instead, they simply know that the i-th action profile played has index i, and if it got eliminated, they can skip it in their count. We are not familiar with any work that has used this simple yet effective observation before. This observation also plays nicely with randomizing a subset of action profiles, used to improve scalability. For the same reason, players do not have to coordinate their randomized subsets.
> >
> > 3. The technical originality stems from having to address the structural subtleties in our proofs. The reviewer is correct that our analysis follows that of MWM. However, applying queuing theory to this non-queuing scenario involves several technical subtleties.
> >
> > - - In particular, while the proof of Lemma 2 (Section 8 in the appendix) is indeed simple, even there we had to take into account the nature of our nontraditional queue. The proof of Theorem 2 however is not simple. It has to incorporate Lemma 3 carefully, which accounts for the learning aspect and is not part of the traditional MWM analysis (even with “standard” queues). It also has to account for our non-standard queues discussed above.
> >
> > - - In the revised version, following the reviewer's comment, we added our recent analysis for time-varying topologies, which adds several new involved technical barriers.
> >
> > - - Note that our Lyapunov function, while indeed being the weighted 2-norm of the system backlog, has very specific weights - the first row of the matrix $W^{D_{E}}$, where $W$ is the communication matrix and $D_{E}$ is the delay parameter. Other weights would not work for our distributed algorithm. For time-varying topologies, the weights of our Lyapunov function are even more subtle, as we explain in the text (Definition 6).
> >
> > **"Can you bring the definitions of strong stability and mean rate stability to Sec. 2?"**
> >
> > We thank the reviewer for spotting this. We moved these definitions to Section 2.
> >
> > **Regarding the definition of the capacity region in Eqn. 2:**
> >
> > - The definition of the capacity region is from the point of view of a centralized algorithm that knows all parameters and can control all players. Indeed, one of the contributions of our paper is to show that this capacity region is achieved with a distributed algorithm as long as the communication graph is connected. Therefore the same capacity region (that centralized algorithms achieve) is achievable for all connected communication graphs. Note that the communication graph, and the algorithm, affect $\varepsilon_{E}$ as given in equation (5), so with harder communication graphs, we will typically need larger $E$ to achieve the same approximate capacity region. A larger $E$ makes the QoS regrets larger as well.  Our new results on time-varying graphs maintain the same capacity region, where now $\varepsilon_{E}$ is given by equation (60).
> >
> > - In case you meant the interaction between the players (e.g., interference in wireless networks), then this is encoded in $\mu_{n}\left(\boldsymbol{a}\right)$ that the capacity region $C_{0}(\mathcal{G})$ depends on, which is a function of the actions of all agents,
> >
> > **"Please correct the LHS of the definition..."**
> >
> > - Thanks. We corrected the notation for $p\left(\boldsymbol{a}\right)$.
> >
> > **"...Shouldn't the RHS of the inequality be "$\geq\lambda_{n}-\varepsilon$" instead of "$>\lambda_{n}+\varepsilon$"?**
> >
> > - It is $>\lambda_{n}+\varepsilon$ since the larger $\varepsilon>0$ is, the smaller the capacity region becomes (includes fewer $\boldsymbol{\lambda}$ vectors). This follows since the approximate capacity region includes only “easier” $\boldsymbol{\lambda}$ which are achievable even by an algorithm that has to learn the stabilizing policy. We use “$>$” instead of "$\geq$” to only consider the interior of this set.
> >
> > **"On line 132, shouldn't that be $r_{n}\left(\boldsymbol{a}\left(t\right)\right)$ ?"**
> >
> > - The reviewer is correct, this is a typo. Thanks!
> >
> > **"Please refer to Lemma 2 somewhere in Lines 138-140"**
> >
> > - This is a good idea. Thanks.

---

> > > ### Author Response · Authors · 2022-08-02
> > > **Response to Reviewer #1: Part 3**
> > >
> > > **Regarding the E tradeoff:**
> > >
> > > It is possible to choose a fixed $E$ with respect to $T$. The tradeoff is between the size of the feasible region and the QoS regret (or delay). For a fixed $E$, only a smaller approximate capacity region $C_{\varepsilon_{E}}\left(\mathcal{G}\right)$ is achievable, where $\varepsilon_{E}>0$ depends on $E$ through equation (5). If we pick $E_{T}$ that increases with $T$, then for large $T$, the full capacity region $C\left(\mathcal{G}\right)$ becomes feasible. The reviewer's hunch is right in the sense that for given QoS $\boldsymbol{\lambda}$, choosing a large enough $E$ such that $\boldsymbol{\lambda}$ is inside $C_{\varepsilon_{E}}\left(\mathcal{G}\right)$ with a margin, but not larger than that is in some sense “optimal” since it minimizes the regret/delay.
> > >
> > > **Time-varying Topologies:**
> > >
> > > We are happy to report that we have just recently completed an analysis of time-varying topologies. Fortunately, OpenReview now allows us to upload a revised version. We included our analysis for time-varying topologies in the revised appendix (Section 12) so that the reviewers can examine it. We will include these new results in the final version if you believe it strengthens the contribution.
> > >
> > > Thanks for pointing out this interesting reference. The paper “Dynamic Server Allocation to Parallel Queues with Randomly Varying Connectivity” by Tassiulas and Ephremides considers a one-hop network with a time-varying connection. For this special case of networks, MWM simplifies into serving the longest queue policy. However, for a general time-varying graph, serving the longest queue is not throughput optimal. This is why our analysis for time-varying topologies still has to follow MWM, even if the “noise” resulting from the varying topology is more intricate. This leads to several technical subtleties we discuss in this new section of the appendix.

---

> > > > ### Comment · Reviewer_1eJN · 2022-08-09
> > > > **Reviewer response**
> > > >
> > > > 1. With regards to the authors' response about "Structural Originality," I'm afraid I remain unconvinced. Departure processes that are determined by other agents in the system are extremely common in random access mechanisms and are well-studied. The same goes for non integral queue lengths.
> > > > 2. I'm not convinced with the argument about the novelty of the Lyapunov function used in Def. 4.
> > > > 3. However, I do agree, as before, that the connection between multi-agent games and MWM is indeed interesting and novel.
> > > > 4. I find the results in Sec. 12 to be quite compelling as well. The fact that SLCQ doesn't necessarily work when the graph isn't fully connected is quite obvious. My intention was to bring to the authors' attention the fact that analogs with time varying graphs do exist.
> > > > * Algorithm 2, its analysis and _this_ Lyapunov function (Eqn. 58) are quite interesting and novel.
> > > >
> > > > In light of this new information, I would like to increase my rating from 6 to 7.

---

### Author Response · Authors · 2022-08-02
**Thank you for all the thoughtful comments!**

We would like to thank all reviewers for their insightful comments that help to improve the quality of the paper. We have responded to all your comments below in detail, and are looking forward to the discussion phase. We would be happy to elaborate on any of these issues further and answer any questions.

For your convenience, **we also uploaded a preliminary revision as the "Supplementary Material"**, where the modified parts are highlighted in blue. This revision includes a generalization to time-varying communication graphs, which appears in the last section of the appendix and requires a new intricate analysis.

---

### Meta-Review · Area_Chair_heVg · 2022-08-26

**Recommendation:** Accept
**Confidence:** Less certain

**Metareview:**

Two of the three reviewers are positive about this work. One reviewer is negative, suggesting that stronger results can be easily derived from the existing literature on matrix games. The authors have replied extensively, explaining why they believe this is not possible, and the same reviewer did not engage their argument in detail. Consequently, I would recommend to accept this paper, albeit the authors should explicitly insert into the main body of the paper the shortcomings of the matrix-game approach.

**Award:**

No

---

### Decision · Program_Chairs · 2022-09-14

Accept